# The Unlearnability Phenomenon in RLVR for Language Models

**Yulin Chen** [1 2]   **He He** [† 1]   **Chen Zhao** [† 1 2]

## Abstract

Reinforcement Learning with Verifiable Reward (RLVR) has proven effective in improving Large Language Model's (LLM) reasoning ability. However, the learning dynamics of RLVR remain underexplored. In this paper, we reveal a counterintuitive phenomenon: among hard examples that the model initially struggles with, a substantial subset remains unlearnable even when correct rollouts are present. To understand the phenomenon, we first demonstrate that existing optimization and sampling techniques fail to resolve unlearnability. With cross-example gradient analysis, we show that unlearnable examples have fundamental representation issue, characterized by low gradient similarity with the rest of the examples and ungeneralizable reasoning patterns. We further show that representation flaws are difficult to mitigate in RL, as data augmentation does not improve gradient similarity. Our study provides the first systematic characterization of unlearnable data in RLVR training and reveals fundamental limitations in current RL approaches for reasoning tasks. Code and data are available at https://github.com/yulinchen99/unlearnability-rlvr.

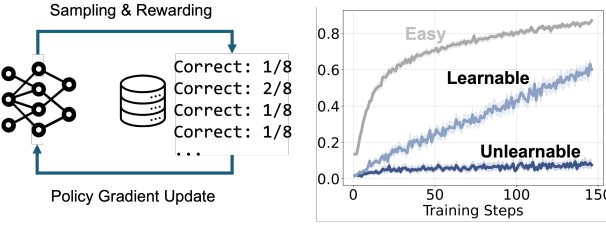

*(a)* Reinforcement learning with verifiable reward (RLVR).

*(b)* Training reward dynamics for different examples.

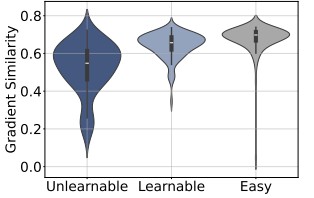

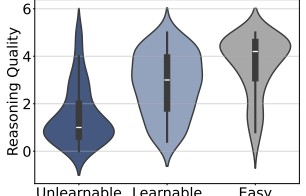

*(c)* Gradient similarity.

*(d)* Reasoning quality.

*Figure 1.* Language models exhibit distinct learning behaviors on difficult examples (i.e., those the model initially struggles with) when trained using reinforcement learning with verifiable reward (RLVR). Measuring improvement via example-level success rates after convergence under GRPO training, we observe that a substantial subset of examples remains unlearned throughout training (Figure 1b). These unlearnable examples exhibit lower gradient similarity to the overall training distribution (Figure 1c) and are often associated with lower-quality reasoning (Figure 1d) compared to both learnable difficult examples and easy examples.

## 1. Introduction

Reinforcement Learning with Verifiable Reward (RLVR) (Shao et al., 2024; Guo et al., 2025) has emerged as the core technique to improve language models' complex reasoning ability, including math (Shao et al., 2024), coding (Hugging Face, 2025; Wei et al., 2025) and agentic tasks (Jin et al., 2025a; Zheng et al., 2025b; Qian et al., 2025), with Group Relative Policy Optimization (GRPO) (Shao et al., 2024) as a standard algorithm.

Intuitively, the success of GRPO relies on the outcome reward variance (Xu et al., 2025) within grouped rollouts, i.e. the existence of both correct rollouts and incorrect rollouts for the same training examples.

While recent work has focused on designing positive rewards for extremely difficult examples (Sun et al., 2025a; Qu et al., 2025), it remains unclear whether the mere presence of positive reward is sufficient for learning. We find that it is not. As shown in Figure 1b, we categorize training examples into three groups: the *easy* group that saturates early in training, the *learnable* group that models initially struggle with but learn smoothly during training, and the *unlearnable* group that consistently receive positive rewards during training yet exhibit no improvement in their reward over time. We refer to this behavior as the *unlearnability phenomenon*, and this paper asks a central question: **why do**

---

[†]Equal advising.

[1]New York University [2]NYU Shanghai. Correspondence to: Yulin Chen <yc7320@nyu.edu>, Chen Zhao <cz1285@nyu.edu>.

*Proceedings of the $43^{rd}$ International Conference on Machine Learning*, Seoul, South Korea. PMLR 306, 2026. Copyright 2026 by the author(s).

**certain examples remain unlearnable despite receiving positive reward signals?**

To investigate this phenomenon, we start with common hypotheses that unlearnability stems from optimization-side issues, including scarcity of positive rollouts, gradient regularization from clipping and KL penalties, or gradient interference between correct and incorrect rollouts. We test each through targeted interventions, including controlling the number of positive rollouts per batch and ablating standard regularization mechanisms. Across all three, the interventions yield no improvement on unlearnable examples. The converging negative results suggest unlearnability is unlikely to be fully explained by standard optimization-side factors, but instead reflecting a fundamental limitation in how models learn from certain types of examples.

We further conduct a deeper analysis of the sampled rollouts during training. Our results indicate that unlearnability stems from flawed internal representations within the language model. Specifically, by computing example-level gradients from positive rollouts, we find that unlearnable examples exhibit substantially lower gradient similarity to the rest of the training data than both easy and learnable examples (Figure 1c). Qualitative inspection of reasoning traces further indicates that although the final answers may be correct, the model frequently produces incoherent or even erroneous intermediate reasoning steps on unlearnable examples (Figure 1d). These representation deficiencies prove difficult to remedy at the RL stage, as neither data augmentation nor curriculum-based training effectively improves gradient similarity or reasoning quality for unlearnable examples. In contrast, we observe that extensive mid-training substantially improves the gradient similarity of difficult examples with the rest of the examples.

Our study reveals the unlearnability phenomenon and performs systematic analysis. Various experiments suggest the LLMs have flawed representations for the unlearnable data that can hardly be fixed during RL post-training stage. We believe the unlearnability phenomenon represents a fundamental limitation of LLM RLVR training.

## 2. Related Works

**Can LLMs Learn New Skills from RLVR.** A large number of works (Yue et al., 2025; Liu et al., 2025a; Wu et al., 2026) that center around understanding RLVR for LLMs aim to answer the question of whether models learn new skills in RL fine-tuning. Starting with the first ever work that shows pass@k degrades after RL (Yue et al., 2025), follow-up works (Yuan et al., 2025; Zhang et al., 2025b) conduct more controlled experiments to explore what exactly the models learn during RL. Some works show that LLMs pick up atomic skills in SFT and learn to compose

them through RL training (Yuan et al., 2025; Park et al., 2025). Others broadly study how well the model can generalize after RL training and how does it relate to initial policy model (Sun et al., 2025a; Zhang et al., 2025b). Wu et al. (2026) provides both theoretical and empirical discussion of why LLMs cannot discover entirely original solutions. Most related to our work, Sun et al. (2025a) studies the learning dynamics of extremely difficult examples with zero initial pass@k, and show models can still successfully learn if fine-grained reward assignment is possible. Our work, on the other hand, challenges the assumption that any examples with positive reward can be learned and show fundamental limitations in RL post-training beyond reward assignment.

**Training Techniques for LLM RLVR.** Since the success of GRPO algorithm, various training techniques have been proposed to improve the original GRPO. The techniques mainly address training efficiency, exploration, or credit assignment problem. DAPO (Yu et al., 2025) proposes clipping higher, and removing KL penalty to encourage LLM explorations. For exploration, existing works often use entropy as an indicator for model exploration and apply entropy-based loss weight adjustment to improve model performance (Cui et al., 2025; Cheng et al., 2025; Jin et al., 2025b). Other works adjust credit assignment by altering the granularity of gradient clipping and optimization (Liu et al., 2025b; Zheng et al., 2025a) to stabilize RL training and improve final performance. In terms of data scheduling design, dynamic sampling (Yu et al., 2025) has been widely applied to improve training efficiency. Meanwhile, curriculum learning, as a more systematic dynamic sampling method, is also shown to improve training efficiency as well (Shi et al., 2025; Gao et al., 2025).

## 3. Unlearnable Examples in LLM RLVR

In this section, we describe the baseline GRPO algorithm we use and provide a working definition of "unlearnable examples" that we adopt to facilitate future analysis.

### 3.1. Training Algorithm

We consider RLVR training with GRPO (Shao et al., 2024) algorithm specifically. The training data contains a set of examples with verifiable answers $\mathcal{D}_{\text{train}} = \{(x, y^*)\}$ and during training $k$ responses are sampled for each example with current policy model $\{y_i\}_{i=1}^k \sim \pi_{\theta_{\text{old}}}(\cdot|x)$. Then the responses $y$ are automatically verified and assigned binary reward $\mathbb{1}[y_i = y^*]$, and the advantage is calculated as the standardized reward $\hat{A}_i = \frac{\mathbb{1}[y_i=y^*]-\text{mean}(\{\mathbb{1}[y_i=y^*]\}_{i=1}^k)}{\text{std}(\{\mathbb{1}[y_i=y^*]\}_{i=1}^k)}$. The policy model is optimized to maximize the PPO (Schulman et al., 2017) loss:

$$\mathcal{L}_{\text{GRPO}}(\theta, (x, y^*)) = -\frac{1}{k} \sum_{i=1}^{k} \frac{1}{|y_i|} \sum_{t=1}^{|y_i|}$$
$$\min \left( r_{i,t} \hat{A}_i, \text{clip}(r_{i,t}, 1 - \varepsilon, 1 + \varepsilon) \hat{A}_i \right) - \beta KL(\pi_\theta | \pi_{\text{ref}}), \quad (1)$$
$$r_{i,t} = \frac{\pi_\theta(y_{i,t}|x, y_{i,<t})}{\pi_{\theta_{\text{old}}}(y_{i,t}|x, y_{i,<t})}$$

To improve training efficiency, we use GRPO with dynamic sampling (Yu et al., 2025) as our baseline RL algorithm, where prompts with zero reward variance are filtered during training. Therefore, at each step, the training loss is:

$$\mathcal{L}_\theta = \mathbb{E}_{(x,y^*) \sim \mathcal{D}_{\text{update}}}[\mathcal{L}_{\text{GRPO}}(\theta, (x, y^*))],$$
$$\mathcal{D}_{\text{update}} = \{(x, y^*) | \text{std}(\{\mathbb{1}[y_i = y^*]\}_{i=1}^{k}) \neq 0\}. \quad (2)$$

### 3.2. Example Learnability

We term all examples with initial success rate $\geq 10\%$ as *easy examples* and others as *hard examples*. Based on the observation in Figure 1b, we can further categorize the hard examples into two **groups**: *learnable examples* $\mathcal{D}_l$ that get improved consistently during training, and *unlearnable examples* $\mathcal{D}_u$ whose reward stays low during training [1]. Note that we have already excluded examples that never observe correct rollouts during training. Our study focuses on the **unlearnable group $\mathcal{D}_u$ with correct rollouts observed**.

**A Working Definition.** To facilitate our study, we provide a working definition of "unlearnable" examples $\mathcal{D}_{\text{train}}$. **The example is considered unlearnable if it does not achieve meaningful improvement in performance when validation performance saturates, despite observing correct samples during training process.** Specifically, we identify unlearnable examples as those with $pass@1 < \tau$ under the final policy, where pass@1 is estimated by sampling $N$ responses per example. We use $\tau = 0.1$ and $N = 32$ across all settings. We also exclude examples that observe no single positive reward throughout RLVR.

### 3.3. Unlearnable Examples Exist Widely

**Experiment Setups** To demonstrate the phenomenon comprehensively, we experiment with Qwen2.5-0.5B (Qwen et al., 2025), Qwen2.5-3B (Qwen et al., 2025), and Llama3.2-3B-Instruct (Grattafiori et al., 2024). We follow previous works that train models on training data with customized difficulty to mimic realistic setups and maximize data utility. Specifically, we train Qwen2.5-0.5B on MATH (Hendrycks et al., 2021) training data from difficulty level 1∼4 (MATH Easy) and Llama3.2-3B-Instruct

on MATH level 3∼5 (Hendrycks et al., 2021) (MATH Hard) as in previous work (Zeng et al., 2025). We use MATH_500 [2] as the validation set. For Qwen2.5-3B, we adopt DeepScaleR (Luo et al., 2025), a large-scale dataset with 40k math problems and verifiable answers. We randomly sample 90% as training set and 10% as validation set. Since the pass@1 threshold is chosen rather arbitrarily, we conduct three independent GRPO trainings for each setting and take the intersection of the different groups of examples as the final subject for analysis to reduce noise. Further training details can be found in Appendix B.1.

**Results** Table 1 presents percentage of unlearnable examples, learnable examples and examples without positive reward throughout training. Across all settings, after excluding data without any positive reward during training, about half the data are learned smoothly while the other half are unlearnable. Overall, we observe that unlearnable data prevails across model and training data settings.

*Table 1.* Percentage of unlearnable examples $\mathcal{D}_u$, learnable examples $\mathcal{D}_l$ and examples with no positive reward during RL training. The percentage is calculated against the number of difficult examples with initial success rate below 0.1. Intersections are taken across three independent trainings for $\mathcal{D}_u$ and $\mathcal{D}_l$, and union set is taken for examples with no positive reward.

| Model | $\mathcal{D}_u$ (%) | $\mathcal{D}_l$ (%) | w/o Pos. Reward (%) |
|---|---|---|---|
| Qwen2.5-0.5B | 30.2 | 25.6 | 23.5 |
| Llama-3.2-3B-Instruct | 21.9 | 31.6 | 37.7 |
| Qwen2.5-3B | 16.7 | 14.2 | 47.2 |

## 4. Examining Common Explanations for Unlearnability

In this section, we explore common hypotheses for the unlearnability phenomenon: (1) scarcity of positive rollouts (Section 4.1), and (2) gradient regularization effect from clipping and KL penalty (Section 4.2) We mainly use Qwen2.5-0.5B trained with MATH Easy dataset and Llama-3.2-3B-Instruct trained with MATH Hard dataset as the concerned settings for analysis. All results presented in this section is for Qwen2.5-0.5B and results for Llama-3.2-3B-Instruct can be found in Appendix A.1.

### 4.1. Positive Rollout Scarcity

Based on previous findings, an intuitive explanation for unlearnability is that the amount of positive rollouts is insufficient for unlearnable examples. Therefore, we have the following hypothesis:

---

[1] When visualizing training rewards for prolonged training steps, some of the unlearnable examples see a sharp increase in reward, accompanied with drop in validation performance.

[2] https://huggingface.co/datasets/HuggingFaceH4/MATH-500

**Oversampling with Replay.** To validate the hypothesis, we apply oversampling with experience replay (Sun et al., 2025b; Zhang et al., 2025d;c) to ensure the ratio of positive samples to negative ones is always the same for each training example. Specifically, we increase the number of sampled rollouts to $4k$ per example and then downsample to $k$, while ensuring each example has exactly $k_{\text{pos}}$ positive rollouts and $k - k_{\text{pos}}$ negative ones in each batch. When there are not enough positive rollouts for the current example, we replay previously sampled positive rollouts from the buffer. Each buffered rollout can be replayed at most two times, and the advantage is calculated after the replay and downsampling process. The detailed algorithm is illustrated in Algorithm 1.

In our experiments, we use $k = 8$ and $k_{\text{pos}} = 1$, that is, for each prompt there is one correct rollout and seven incorrect ones participating in gradient calculation and policy optimization. We focus on this setting because the replay rate is already high for the unlearnable examples and the sampling time cost is large when $k$ scales up.

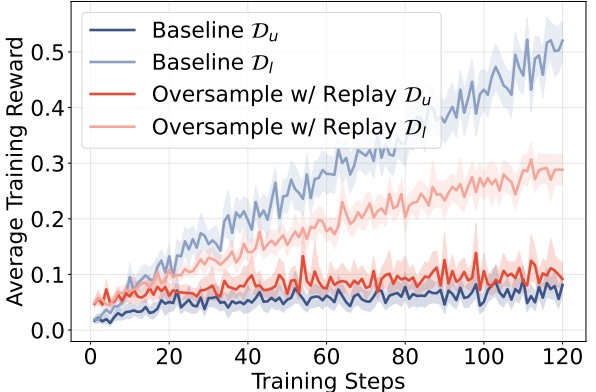

*Figure 2.* Comparison of training reward dynamics on learnable $\mathcal{D}_l$ and unlearnable groups $\mathcal{D}_u$ for naive GRPO and GRPO with oversampling and replay method. The figure shows the gap between learnable and unlearnable examples cannot be fully attributed to difference in number of correct rollouts.

**Results.** The training reward curve after applying the oversampling is shown in Figure 2.[3] Controlling the number of correct rollouts effectively slows down the learning pace of learnable data. However, it does not resolve the issue of un-

---

[3]Note that although the rollouts may be discarded or replayed for optimization purpose, the training reward shown in Figure 2 is calculated as the average reward of the original $4K$ samples for each prompt. To make a fairer comparison, we also exclude prompts that get filtered before gradient descent due to absence of correct rollout in both current sampling batch and buffered rollouts.

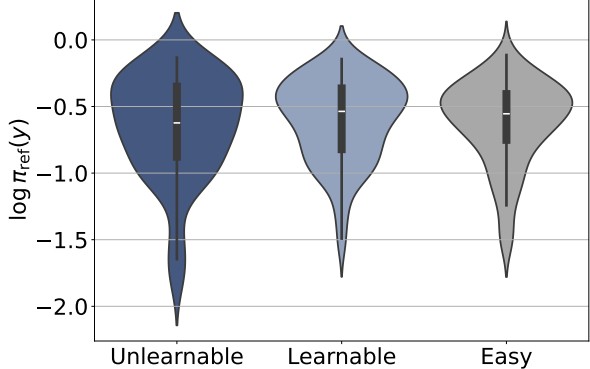

*Figure 3.* Distribution of reference log-likelihood for different data examples' correct rollouts. All rollouts are sampled from the initial policy model. Unlearnable examples do not necessarily have low-probability rollouts.

learnability, and the gap between learnable and unlearnable groups remain. We further verify in Appendix A.2 that the gap persists under two stronger interventions: supervised fine-tuning on distilled correct responses and RL with a substantially larger rollout group ($k = 64$) on unlearnable examples alone. Neither closes the gap, indicating that unlearnability is not resolved by more positive rollouts. This result indicates that the gap is not an issue of lack of positive reward signals, but rather reveals more fundamental difference between the two groups of data.

### 4.2. Gradient Regularization

Standard RL methods often incorporate constraints to ensure stable training. Clipping mechanisms (Schulman et al., 2017) suppress gradients for low-probability tokens, while KL loss term (Schulman et al., 2017) penalizes deviation from a reference model. Both mechanisms can wash out the positive signal from correct rollouts before it influences learning, and some existing works (Yu et al., 2025; Yue et al., 2025) also show that clipping higher and removing the KL penalty term can improve model performance after RLVR. Therefore, we have the following hypothesis:

**Reference probability of correct rollouts.** We sample correct rollouts from the initial policy and measure their reference log-likelihood across the three groups. As shown in Figure 3, the distribution is comparable for unlearnable, learnable, and easy examples, with no systematic shift to-

ward lower probabilities for the unlearnable group.

**Clipping rates during training.** A second implication is that unlearnable examples should incur higher clipping rates in practice. Figure 4 shows the realized clipping ratio across the three groups over the course of training. The curves track each other closely, indicating that unlearnable examples are not disproportionately clipped.

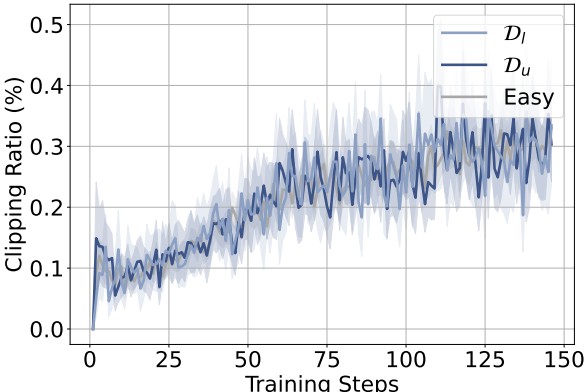

*Figure 4.* Comparison of clipping ratio for different groups of examples. The clipping effect on all three groups is similar.

**Ablating clipping and KL regularization.** If clipping or KL constraints were responsible for the lack of learning on unlearnable examples, relaxing them should benefit unlearnable examples. We train with clip-higher (Yu et al., 2025) and with the KL term removed. Figure 5 shows neither intervention changes the training dynamics on unlearnable examples, and the gap between learnable and unlearnable groups persists at essentially the same magnitude as in the baseline.

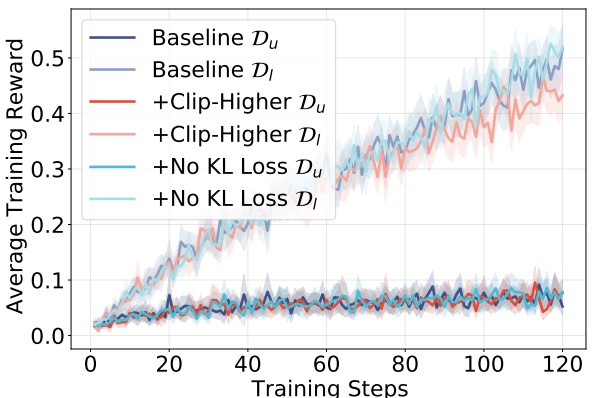

*Figure 5.* Comparison of training reward dynamics on learnable $\mathcal{D}_l$ and unlearnable groups $\mathcal{D}_u$ for naive GRPO, GRPO with higher clipping, and GRPO without KL loss term. Both training techniques do not improve learnability.

This finding indicates that unlearnable examples are not

edge cases affected by clipping mechanisms or KL divergence constraints, and that their resistance to learning stems from factors beyond low initial probabilities under the reference policy. More analysis results on gradient interference (Nguyen et al., 2025) can be found in Appendix A.3.

## 5. Unlearnability Suggests Representation Issue

Section 4 examines three natural hypotheses for failure in RL and shows the unlearnability phenomenon does not stem from data imbalance or optimization mechanics. This suggests that unlearnability is not an artifact of the RL training, but rather reflects something more fundamental about the interaction between certain examples and models. In this section, we examine the position of unlearnable examples in the optimization space through cross-prompt gradient analysis and conduct reasoning quality analysis on the "correct" rollouts. All experiments in this section are performed with Qwen2.5-0.5B, and some key findings are also reported for Llama-3.2-3B-Instruct in Appendix A.1.

### 5.1. Unlearnable Examples Are Gradient Outliers

**Computing cross-example gradient similarity.** Example-level gradients serve as a more direct proxy for training dynamics. To calculate gradient for each example, we sample 100 examples from each group and 1000 rollouts per example under the initial policy, filter for the correct rollouts, and compute the GRPO loss following Equation 1. The per-rollout gradient is averaged first across tokens within the response and then across responses, yielding one gradient vector per example for each label. Then we obtain the cosine similarity between gradients of each pair of examples. For computational efficiency, we attach a fixed, randomly initialized LoRA adapter and compute gradients with respect to LoRA parameters only. On the 0.5B model, LoRA-based gradient similarity is highly correlated with full-parameter gradient similarity.

**Results.** Figure 1c shows the distribution of the average gradient similarity for different groups. It can be seen unlearnable examples have much lower gradient similarity with the rest of the examples. This is a direct evidence that what the model learns in the other two groups do not transfer to the unlearnable group, and also explains why the reward gap still exists after controlling the number of positive rollouts in Section 4.

We also report the inter-group and intra-group gradient similarity in Figure 6. Surprisingly, easy examples seem to share highly consistent gradients, while unlearnable examples are not similar to any other groups. This further suggests that each individual example in the unlearnable group is an outlier in the gradient space, whereas the learning signals are

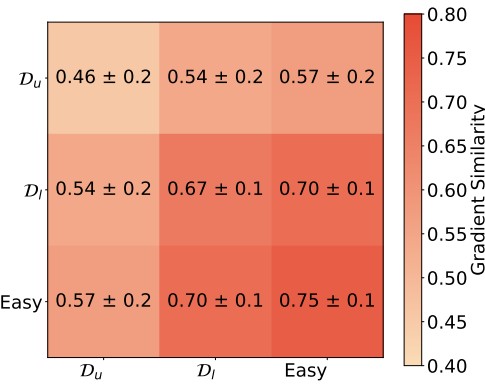

*Figure 6.* Gradient similarity of correct rollouts across unlearnable, learnable, and easy training examples. The rollouts are sampled with the initial policy model and the gradients are calculated against the same model. Easy examples have highly concentrated gradients while unlearnable examples are distinct gradient outliers.

highly aligned for easy examples.

Figure 17 in Appendix A.4 shows the gradient similarity distribution calculated at step 50 midway through the RL training. The overall gradients are more spread out over the optimization space as a result of model update and the gradient similarity of unlearnable examples remains low. The consistently low gradient similarity for the unlearnable group implies uniformly weak skill transferability from broader training data to the unlearnable examples. Overall, we observe correlation between gradient similarity and learnability during RLVR training, and the fact that unlearnable examples are gradient outliers indicates that models have *flawed representations* for unlearnable examples.

### 5.2. Unlearnable Examples Show Ungeneralizable Reasoning Patterns

Since RLVR assigns rewards solely based on final answer correctness, we also analyze the quality of reasoning traces for different examples. We randomly sample 100 examples from each group and gather their rollouts with correct final answer. We prompt GPT-5-mini (Singh et al., 2025) to generate quality score from 0 to 5.

The results are shown in Figure 1d. Even though all responses labeled arrive at the correct answer, the quality of reasoning is correlated with the initial success rate, where model generates higher-quality reasoning for the easy examples. Comparing unlearnable examples with learnable ones, model produces substantially lower-quality reasoning on unlearnable examples at initialization. Table 2 presents an example low-quality reasoning trace. The model starts with correct analysis but makes some serious mistakes in the subsequent case enumeration. The last part of the reasoning also shows inconsistency with its own analysis in the beginning, deviating from the original problem. The

fact that the flawed reasoning leads to a final correct answer indicates the model is not actually "reasoning", but rather exploiting some ungeneralizable shortcut solution or bag of heuristics (Nikankin et al., 2025). This also points to the limitation of solely using outcome reward in RLVR without validating intermediate reasoning steps, where the model unintentionally "hacks" the reward with "fake reasoning" and makes the training signals noisy.

Then we investigate whether the reasoning quality improves during RLVR training. Figure 7 shows the gap between unlearnable and learnable examples gets even larger as the training proceeds. The reasoning quality on learnable ones improves substantially in early stage of training, while the effect does not transfer well to the unlearnable data and the reasoning quality remains low for many examples. Experiments with curriculum learning in Appendix A.5 further confirm the finding. Even when training only on easy and learnable examples, reasoning quality on unlearnable examples fails to improve. The persistently low reasoning quality provides further evidence that models rely on ungeneralizable reasoning patterns to achieve correct answers on these examples, which is another sign of flawed representation.

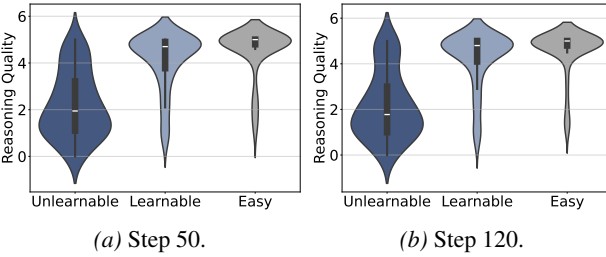

*(a)* Step 50.      *(b)* Step 120.

*Figure 7.* Distribution of model response reasoning quality for examples in different groups during training. Scores are generated by GPT-5-mini on sampled responses with correct final answers. The reasoning quality on unlearnable examples fails to improve effectively during training.

### 5.3. Data Augmentation Does Not Improve Gradient Similarity

**Data Augmentation.** We then explore whether data with high gradient similarity can be synthesized. Intuitively, learning to solve problems with similar strategy and subproblems should benefit learning to solve the original problems. Therefore, we devise two approaches to generate augmented training data: (1) generate **similar problems** $\mathcal{D}_u^{sim}$ and (2) generate **subproblems** $\mathcal{D}_u^{sub}$. For each example, we prompt GPT-5 (Singh et al., 2025) to generate 5 similar problems that can be solved with the same strategy. We also prompt GPT-5 to decompose and generate subproblems such that solving the subproblems would readily give the answer of the original problem. For each augmented problem, GPT-5 is also required to provide an answer and a step-by-step solution at the same time. The generated answers are cross-

*Table 2.* An example of low-quality reasoning trace. Flawed reasoning is in red.

| Problem |
| --- |
| Find the volume of the region in space defined by $\|x + y + z\| + \|x + y - z\| \leq 8$ and $x, y, z \geq 0$. |

| Reasoning |
| --- |
| To find the volume of the region in space defined by $\|x + y + z\| + \|x + y - z\| \leq 8$ and $x, y, z \geq 0$, we need to carefully analyze the given inequality. The regions defined by this inequality can be described by a square in the first octant (where $x, y, z \geq 0$), considering the constraints. ...... **Case 2:** $\|x + y + z\| = 0$ This means $x + y + z = 0$. $\|x + y - z\| = 8$ $x + y - z = \pm 8$ (i.e., $x + y = z \pm 8$). Since $x, y, z \geq 0$, the only valid solution is $x + y = z$ or $z = 0$. **Case 3:** $\|x + y - z\| = 0$ This means $x + y - z = 0$. $\|x + y + z\| = 8$ $x + y + z = \pm 8$. Since $x, y, z \geq 0$, the only valid solution is $x + y + z = 8$ or $0$. So, the region in the first octant where the inequality $\|x + y + z\| + \|x + y - z\| \leq 8$ holds is a square with vertices $(0, 0, 8)$, $(0, 0, -8)$, $(0, 8, 0)$, and $(0, -8, 0)$. The area of this square is $(8 - (-8))^2 = 16^2 = 256$. The volume of the region is the area of the square multiplied by the area of the unit cell in 3D (which is $\frac{1}{8}$ of the area of the unit cube): Volume $= 256 \times \frac{1}{8} = 32$ Therefore, the volume of the region is $\boxed{32}$. |

validated by Gemini-2.5-pro to ensure correctness. We run the data augmentation pipeline for all examples in the unlearnable group to facilitate training. We also synthesize augmented data for 100 random examples sampled from the learnable group for comparative analysis. The details on data augmentation can be found in Appendix C.

**Setups.** We adopt three different data compositions for RL fine-tuning: (1) the original unlearnable examples combined with the similar problems $\mathcal{D}_u + \mathcal{D}_u^{sim}$; (2) the original unlearnable examples combined with the subproblems $\mathcal{D}_u + \mathcal{D}_u^{sub}$; and (3) the combined augmented set $\mathcal{D}_u + \mathcal{D}_u^{sim} + \mathcal{D}_u^{sub}$. We train the model with the unlearnable examples along with its augmented data using the same GRPO algorithm. All training hyperparameters are the same as the baseline trained on original full data $\mathcal{D}$.

**Results.** As shown in Figure 8, all augmented data are of limited help with the unlearnability phenomenon. Even though some slight improvements are visible after 60 steps of training, the evaluation pass@1 performance saturates at very early stage of training and pass@16 score drops after 20 steps of training. Therefore the improvement on training examples is likely overfitting on the unlearnable examples after repeated training on the same set of examples. Figure 9 further reveals the learnability of the augmented data compared with the reward dynamics of original easy

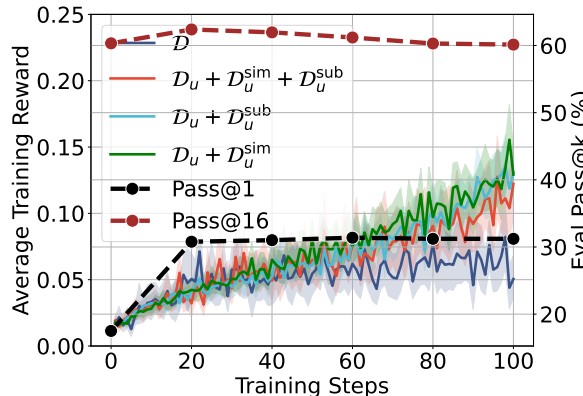

*Figure 8.* Comparison of training rewards on original unlearnable group across different training data compositions. The y-axis on the right shows the pass@k performance on the validation set for the combined augmented setting. Data augmentation does not help model learn the unlearnable examples.

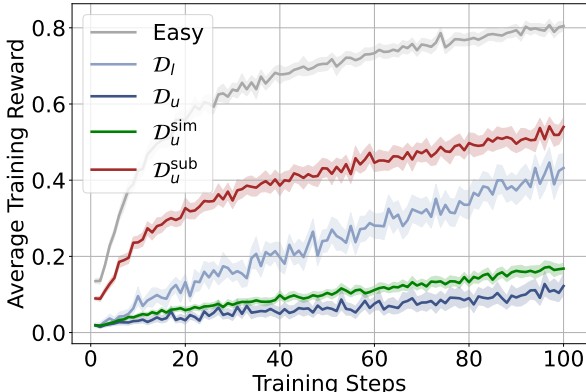

*Figure 9.* Evolution of training reward for the original easy examples, learnable examples $\mathcal{D}_l$, unlearnable examples $\mathcal{D}_u$ and the augmented data, including similar problems $\mathcal{D}_u^{sim}$, and the augmented subproblems $\mathcal{D}_u^{sub}$. The training reward for $\mathcal{D}_u$, $\mathcal{D}_u^{sim}$, and $\mathcal{D}_u^{sub}$ are extracted from the combined augmented setting. The training reward for easy examples and learnable examples are extracted from baseline setting. Even though models may learn smoothly on the augmented data, they still fails to transfer to learning the original unlearnable examples.

and learnable examples. It is worth noting that although the augmented subproblems show even better learnability than the original learnable examples, the effect on unlearnable examples is still limited. This indicates that RLVR fails to incentivize model's ability to compose skills for certain examples.

We further check the gradient similarity between the augmented similar examples and original examples. Surprisingly, although they almost share identical problem structures, the gradient differs greatly. For each unlearnable and learnable example, we plot gradient similarity with the original training set against similarity with the correspond-

ing augmented examples as in Figure 10. In the learnable group, gradient similarity with augmented data is substantially higher than that of the unlearnable group. Moreover, the unlearnable examples exhibit high correlation between the two similarity measures, suggesting that the original unlearnable examples are inherently distinct in gradient space. These findings reveal that semantically similar examples are not necessarily similar in optimization space, and thus may not help the model acquire the same skills during training. This points to an unexpected non-triviality in data synthesis for LLM post-training, particularly for examples that models initially struggle with.

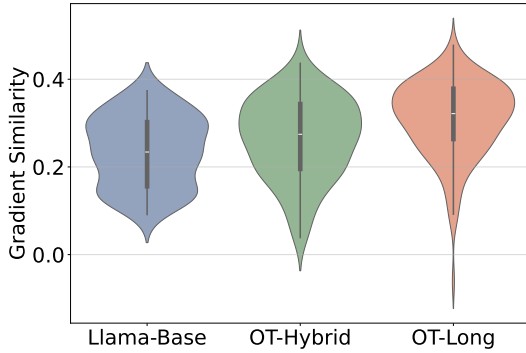

*Figure 11.* Gradient similarity of difficult examples with the broader training distribution for Llama-3.2-3B-base, OctoThinker-3B-Hybrid-Base, and OctoThinker-3B-Long-Base (Wang et al., 2025). Mid-training is shown to improve gradient similarity effectively.

## 6. Discussion

**Implications on Understanding of RLVR.** Our study engages with the existing works on uncovering limitations of RLVR for LLMs. We offer a unique perspective that focuses on the difficult subset of training data and show that a number of unlearnable examples persist given specific data and model setups, despite having correct rollouts during training. The unlearnability is a counterintuitive phenomenon for RL training and further rollout gradient analysis suggests that the models inherently have flawed representation for the unlearnable examples. The unlearnable examples are consistent gradient outliers in the optimization space that are hardly resolved by data augmentation. This view also aligns with recent evidence that reasoning models' internal representations carry rich latent signals (Zhang et al., 2025a) that are learned while acquiring reasoning ability. Our results trace a failure mode in the opposite direction. When the underlying representation is flawed to begin with, outcome-based RL has no clear path to repair it, and the example persists as a gradient outlier regardless of how many correct rollouts are sampled. This finding indicates that positive reward alone does not necessarily lead to smooth model learning. It also highlights the important role of sampling rollouts for training examples, an aspect often overlooked in existing RL training pipelines. Identifying which rollout features beyond outcome correctness contribute to smooth and generalizable LLM optimization represents a promising direction for future research.

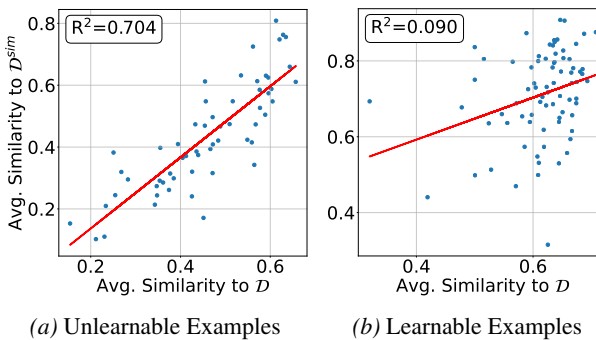

| (a) Unlearnable Examples | (b) Learnable Examples |

*Figure 10.* The correlation of gradient similarity with augmented similar data and with broader training data, for unlearnable and learnable examples respectively. Gradient similarity with unlearnable examples is highly correlated meanwhile also lower than learnable examples.

### 5.4. Mid-training Can Increase Gradient Similarity

Mid-training has shown to be effective to improve base model to make it more suitable for RL stage (Wang et al., 2025). Motivated by this observation, we examine whether mid-training alleviates the unlearnability phenomenon by improving representation alignment prior to RL. We conduct gradient analysis on OctoThinker-3B-Hybrid-Base and OctoThinker-3B-Long-Base (Wang et al., 2025), comparing them to Llama-3.2-3B-Base. The OctoThinker models are mid-trained from Llama-3B on 20B tokens with different data mixtures. For each model, we randomly sample 500 examples from the MATH Hard training set and identify a subset of difficult examples. We then compute the gradient similarity between each difficult example and the full set of 500 examples. According to Figure 11, both mid-trained OctoThinker models exhibit consistently higher gradient similarity than the base Llama model. This indicates that mid-training substantially improves representation alignment for difficult examples, by reshaping model representations before reinforcement learning, rather than relying on RL alone to correct misaligned or flawed representations.

**Implications for Reasoning LLM Training Pipelines.** Our analysis reveals that unlearnability is fundamentally a representation issue that is difficult to address directly at the RL stage. The pre-trained representations for these hard examples appear insufficiently structured to support effective

and generalizable learning. In contrast, mid-training (Wang et al., 2025; Zhang et al., 2025b) appears to reshape the representation space in ways that improve gradient similarity on hard examples, while also producing a more suitable initial policy model that benefits substantially more from subsequent RL fine-tuning. These findings highlight the critical importance of the mid-training stage in the reasoning LLM training pipeline. However, several key questions remain open: what data are most effective for mid-training, and which algorithms best facilitate mid-training.

**Limitations.** Due to resource limit, our experiments are conducted on small to mid-scale base and instruct models in the mathematical reasoning domain. Our working definition of unlearnability relies on a thresholded notion of convergence and is therefore a useful operational tool rather than a sharp categorical distinction, and a fraction of examples will sit near the boundary under any such criterion.

**Conclusion.** In this study, we show some training examples cannot be learned even when correct rollouts are sampled during LLM RLVR training, and that this does not result from lack of positive rollouts or imperfect exploration in RL. Through gradient analysis, we reveal the unlearnable examples are gradient outliers in the optimization space and likely have incoherent reasoning traces. We also show data augmentation and curriculum learning fail to improve gradient similarity or reasoning quality during RL training, suggesting that unlearnability reflects a fundamental flaw in model representations that is difficult to resolve through RL post-training. In contrast, mid-training effectively improves gradient similarity. Our study highlights an overlooked fundamental limitation in RL post-training for reasoning.

## Acknowledgements

We thank Pavel Izmailov, Zayne Sprague, Jingyan Shen, Yunzhen Feng, and Nicholas Lourie for their constructive feedback while developing this project. This work was supported by Global AI Frontier Lab and Coefficient Giving (Open Philanthropy). This work was also supported in part through the NYU IT High Performance Computing resources, services, and staff expertise.

## Impact Statement

This paper presents work whose goal is to advance the field of Machine Learning. There are many potential societal consequences of our work, none which we feel must be specifically highlighted here.

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

# A. Additional Results

## A.1. Results on Llama-3.2-3B-Instruct

We conduct similar analysis for Llama-3.2-3B-Instruct model with its corresponding training data. Figure 12 demonstrates that common hypotheses about positive rollout scarcity and gradient clipping effect do not hold on Llama-3B model as well. Figure 13 presents gradient similarity distribution, where unlearnable examples are more likely to be outliers in the optimization space. And Figure 14 shows data augmentation does not help model learn the unlearnable examples effectively, suggesting the difficulty to fix unlearnability issue in RL post-training stage.

It should also be noted that Llama model seems to have overall less aligned gradient compared to Qwen. Whereas easy examples for Qwen model have highly concentrated gradients with cosine similarity above 0.7, the cosine similarity of gradients for Llama is largely below 0.6 and distribution is much flatter. We believe this partly explains why Qwen is shown to be more suitable for RL training.

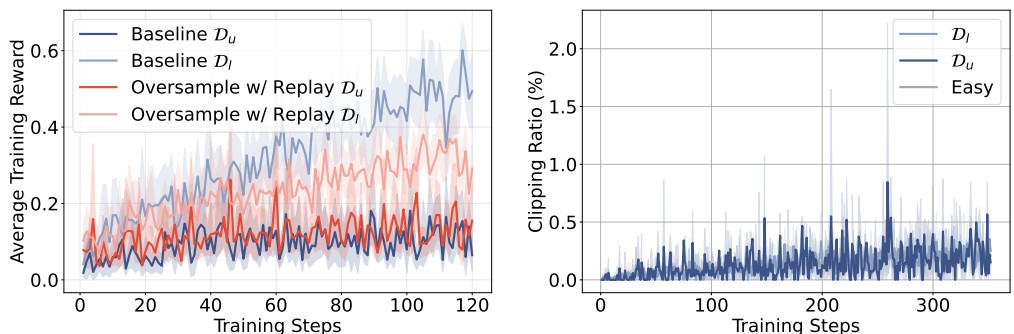

*(a)* Training reward curve for positive rollouts control.*(b)* Clip ratio for different examples throughout training.

*Figure 12.* Analysis results for controling positive rollout number and gradient clipping effects for Llama-3.2-3B-Instruct.

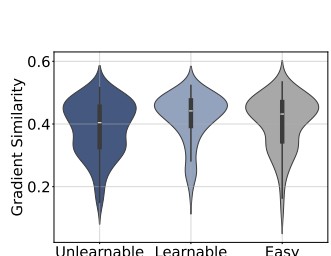

*(a)* Distribution of gradient similarity with the rest of the data for different examples.

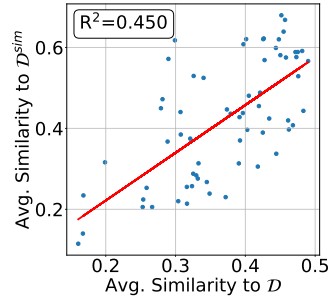

*(b)* Gradient similarity with the augmented data against with other training data for unlearnable examples.

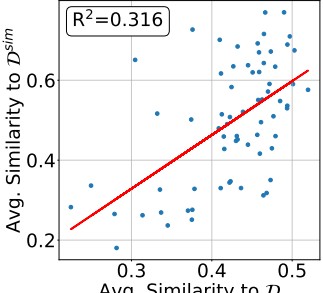

*(c)* Gradient similarity with the augmented data against with other training data for learnable examples.

*Figure 13.* Cross-example gradient analysis results for Llama-3.2-3B-Instruct. Unlearnable examples have lower gradient similarity with the rest of the examples and with the augmented data.

## A.2. Further Results on Positive Rollout Scarcity Hypothsis

To strengthen the case against positive rollout scarcity, we conduct two additional experiments that go beyond the $k_{pos} = 1$ setting in Section 4.1.

**SFT on filtered correct responses.** We generate responses on the MATH training set with Qwen2.5-7B, filter for correct ones, and construct an 800-example SFT dataset comprising 200 unlearnable, 200 learnable, and 400 random easy examples. We then fine-tune Qwen2.5-0.5B on this set and evaluate the per-group pass rate at each checkpoint (Figure 15a). Even

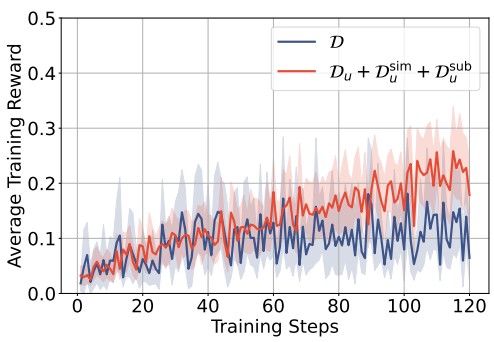

*(a)* Training reward curve for data augmentation experiments.

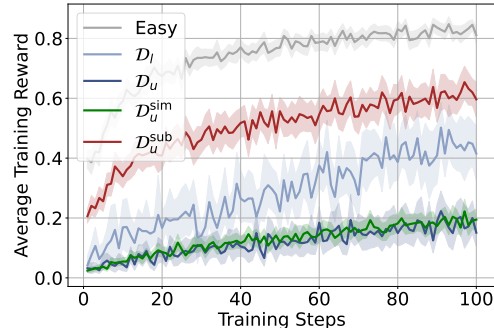

*(b)* Training rewards for original examples and augmented examples.

*Figure 14.* Data augmentation results for Llama-3.2-3B-Instruct. Although models learn augmented subproblems successfully, it does not transfer to improvements in the original unlearnable examples.

under direct supervision on correct responses, which is a substantially stronger signal than RLVR, unlearnable examples remain resistant to learning, while learnable and easy examples improve as expected.

**Larger rollout group with unlearnable-only training.** We further increase the rollout group size to 64 and run RL training on unlearnable examples alone (Figure 15b). Despite a much larger budget of correct rollouts per gradient step and many training epochs, the average reward does not improve meaningfully.

Together, these results indicate that the resistance of unlearnable examples to learning is not closed by either denser positive rollouts or stronger forms of supervision, providing further evidence against the hypothesis.

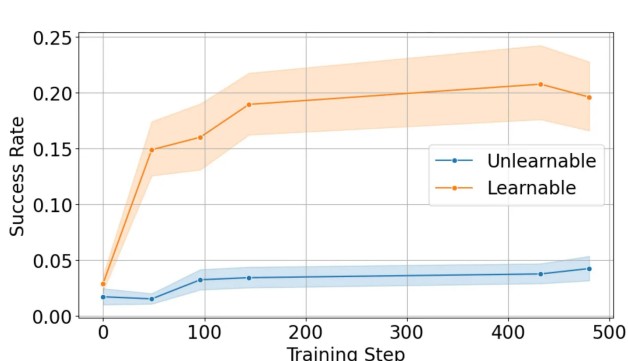

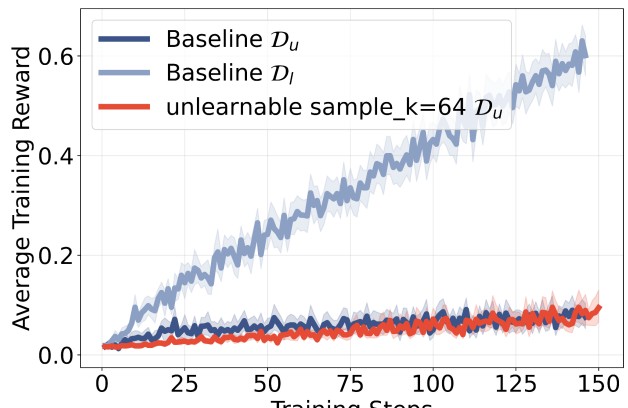

*(a)* Pass rate over SFT checkpoints. Qwen2.5-0.5B is fine-tuned on 800 correct responses generated by Qwen2.5-7B (200 unlearnable, 200 learnable, 400 easy). Even under direct supervision, unlearnable examples remain resistant to learning, while learnable and easy examples improve as expected.

*(b)* Average training reward when running GRPO on unlearnable examples only with rollout group size $k = 64$. Despite a much larger budget of correct rollouts per gradient step and many training epochs, the reward does not improve meaningfully.

*Figure 15.* Additional results for positive rollout scarcity hypothesis. Neither stronger supervision via SFT (left) nor a substantially larger rollout group under RL (right) closes the gap between unlearnable examples and the rest of the data.

### A.3. Hypothesis 3: Gradient Interference

The success of RL depends on exploiting both correct and incorrect rollouts. In LLM RLVR, the gradient is calculated at token level while the reward and advantage is calculated at rollout level. This mismatch in granularity results in imprecise credit assignment. A correct rollout may fail to be learned if it shares critical tokens with incorrect rollouts, causing their gradients to interfere during optimization (Nguyen et al., 2025). This effect is amplified when correct rollouts are outnumbered by incorrect ones. Therefore, we have the following hypothesis:

**Hypothesis 3**

Within the rollout group of an unlearnable example, gradients from correct rollouts are cancelled by gradients from incorrect rollouts that share critical tokens, leaving little net learning signal.

**Within-prompt analysis.** We first measure cosine similarity between gradients of correct and incorrect rollouts within the same example. A cosine similarity score $< 0$ indicates potential gradient interference effects. Figure 16a shows that at the initial policy, gradients from a single example's correct and incorrect rollouts are highly similar across all hard examples, with no distinction between learnable and unlearnable data. As training proceeds, gradients on incorrect rollouts from learnable examples shift to oppose the correct ones, mimicking the distribution of easy examples. However, the similarity distribution for unlearnable examples remains unchanged, with correct and incorrect rollouts staying aligned in optimization space rather than interfering.

**Cross-prompt analysis.** We further examine potential interference effects from all incorrect rollouts in the batch, not just those within the same example. Figure 16b shows no clear difference between learnable and unlearnable groups at this dataset level either.

**Takeaway.** Overall, we observe little gradient interference effect at the rollout level, whether measured within a single example or across the full training batch. The failure to learn on unlearnable examples cannot be attributed to gradients from correct rollouts being cancelled by those from incorrect ones.

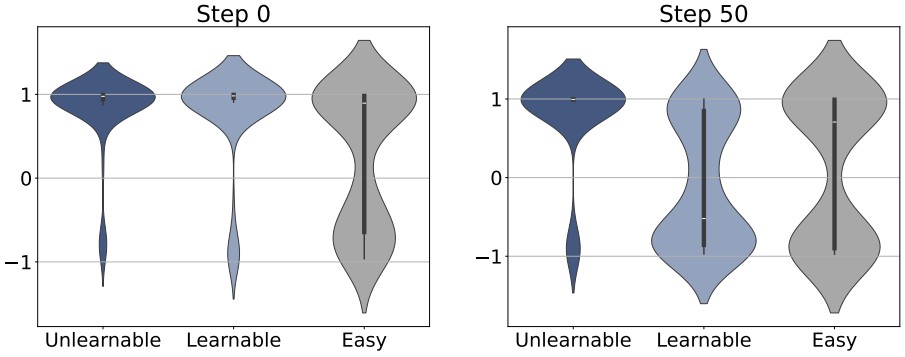

*(a)* Gradient similarity of correct and incorrect rollouts for each individual example.

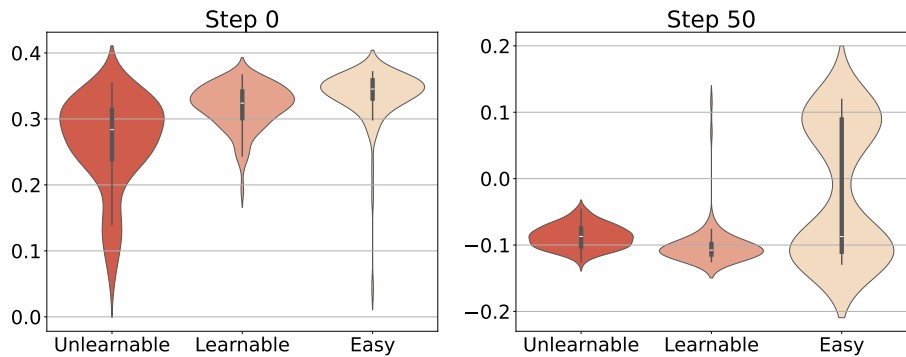

*(b)* Gradient similarity of correct rollouts from each group with incorrect rollouts from the full training data.

*Figure 16.* Gradient similarity between correct and incorrect rollouts during RL training. The similarity score is measured by cosine similarity; a score $< 0$ indicates potential gradient interference effects. Figure a shows within-prompt similarity and Figure b shows dataset-level similarity. Across both views, gradients on unlearnable examples remain aligned rather than interfering.

## A.4. Cross-Example Gradient Similarity During Training

Figure 17 shows the gradient similarity distribution midway through the training for different groups. It can be seen that the distribution for easy data changes drastically as an effect of model optimization. Whereas the gradient similarity for unlearnable examples remain lower than the learnable counterparts.

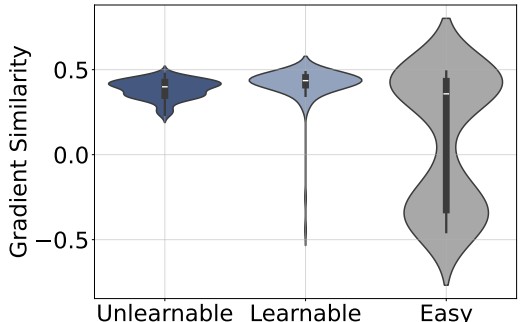

*Figure 17.* Gradient similarity calculated with model after training for 50 steps.

## A.5. Curriculum Learning Does Not Improve Reasoning Quality

Curriculum learning (Bengio et al., 2009) has also been shown to make RLVR more efficient (Shi et al., 2025; Gao et al., 2025). Given above observation on reasoning quality, a straightforward hypothesis is that curriculum learning may also improve reasoning quality by only training on high-quality rollouts from easier examples at each stage.

**Setups.**    Much similar to curriculum learning, in the first stage, we exclude unlearnable examples from the training set and train models until convergence. For second stage, we randomly sample same number of examples as the unlearnable group from first-stage training data and combine with the unlearnable part. We continue to train the model on the new training set.

**Results.**    Figure 18a and Figure 18b show that training on learnable data first does not boost reasoning quality on unlearnable data, nor does it improve success rate. Even though the model improves substantially after first stage training, it does not show benefit when it continues to train in the second stage. In fact, we also observe an obvious drop in validation accuracy when switching to the second stage of training. This result further validates the findings that the learning transferability to the unlearnable data is weak.

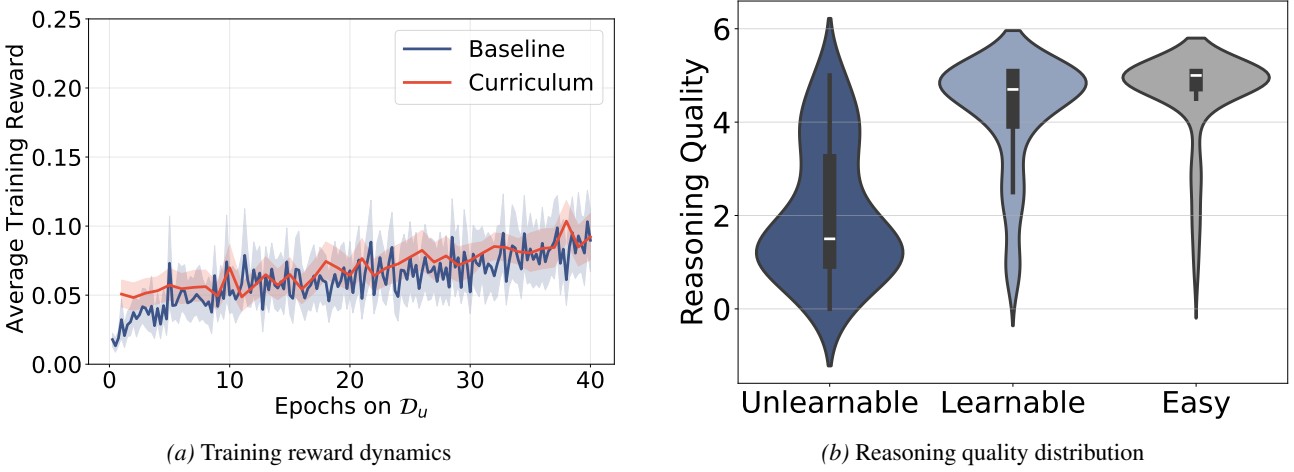

*(a)* Training reward dynamics                    *(b)* Reasoning quality distribution

*Figure 18.* Analysis of curriculum training. (a) Comparison of the training reward dynamics of unlearnable group for baseline full-data training and data curriculum. Due to inconsistency in data schedules, the x-axis represents the number of epochs the unlearnable examples are trained with. (b) Distribution of reasoning quality for examples in different groups after first stage of curriculum training. Scores are generated by GPT-5-mini on sampled responses with correct final answers.

# B. Experimental Details

## B.1. RLVR Training Details

We use verl [4] for full-parameter RL fine-tuning. We use sampling batch size as 256 examples for Qwen2.5-3B and Llama-3.2-3B-Instruct, and 1024 for Qwen2.5-0.5B. The default rollout group size is 8 for each example and the batch size for gradient step update is 64, meaning that the total batch size calculated at rollout level for each gradient update is 512. We use learning rate 5e-7 for Qwen2.5-0.5B and LLama-3.2-3B-Instruct, and learning rate 1e-6 for Qwen2.5-3B. We have also tried different sampling batch size and gradient update batch size to vary the maximum number of off-policy update. The results show no significant difference in unlearnable subset of examples. For inference and sampling, we use temperature 1.0 and maximum length as 5120 for Qwen2.5-05B and Llama-3.2-3B-Instruct and 8192 for Qwen2.5-3B. For answer verification, we use Math-Verify [5].

## B.2. Algorithm for Oversampling with Rollout Replay

Algorithm 1 shows the specific sampling intervention we apply to standard GRPO training in Section 4.1.

---

**Algorithm 1** GRPO with Oversampling and Replay Buffer

---

**Require:** $\pi_\theta$, $\pi_{\text{ref}}$, $\mathcal{D}_{\text{train}}$, batch size $N$, rollouts per prompt $k$, positive samples per prompt $k_{\text{pos}}$, learning rate $\alpha$

  Initialize replay buffer $\mathcal{B} \leftarrow \emptyset$

  **for** each training iteration **do**

    Sample batch of examples $\{(x_i, y_i^*)\}_{i=1}^N \sim \mathcal{D}$

    **for** each prompt $x_i$ in batch **do**

      $\{y_i^{(j)}\}_{j=1}^{4k} \sim \pi_\theta(\cdot|x_i)$         {sampling}

      $r_i^{(j)} = \mathbb{1}[y_i = y^*]$ for $j = 1, \ldots, 4k$

      $\mathcal{P}_i = \{y_i^{(j)} : r_i^{(j)} = 1\}$

      $\mathcal{N}_i = \{y_i^{(j)} : r_i^{(j)} = 0\}$      {gather positive and negative rollouts}

      $\mathcal{B}.add(\mathcal{P}_i, x_i)$     {buffer}

      **if** $|\mathcal{P}_i| \geq k_{\text{pos}}$ **then**

        $\mathcal{P}_i \leftarrow sample(\mathcal{P}_i, k_{\text{pos}})$

      **else**

        $k' \leftarrow k - |\mathcal{P}_i|$

        $\mathcal{P}_i \leftarrow sample(\mathcal{B}, x_i, k') \cup \mathcal{P}_i$

      **end if**

      $\mathcal{N}_i \leftarrow sample(\mathcal{N}_i, k - k_{\text{pos}})$

      $\mathcal{S}_i \leftarrow \mathcal{P}_i \cup \mathcal{N}_i$

    **end for**

    **Compute advantages:**

    **for** each prompt $x_i$ **do**

      $\hat{A}_i^{(j)} = \frac{r_i^{(j)} - \frac{1}{k}\sum_{j=1}^k r_i^{(j)}}{std(\{r_i^{(j)}\}_{j=1}^k)}$ for $y_i^{(j)} \in \mathcal{S}_i$

    **end for**

    Update policy: $\theta \leftarrow \theta - \alpha\nabla_\theta \mathcal{L}_\theta(\pi_{\text{ref}}, \pi_\theta, x_i, \hat{A}_i^{(j)}, \mathcal{S}_i)$

  **end for**

---

[4]https://github.com/volcengine/verl
[5]https://github.com/huggingface/Math-Verify

# C. Data Augmentation Details

## C.1. Prompts

---

Prompt for Synthesizing Similar Problems

Generate 5 new problems that test the same core skills and reasoning patterns as the problem below. The new problems should have similar difficulty and structure but use different contexts, numbers, or scenarios.
Original problem: {problem}
Requirements:
- Each new problem must be solvable and well-defined
- The answer must be a mathematical expression or a number
- Vary the surface details (context, numbers, names) while keeping the underlying logic consistent
- Maintain similar complexity and difficulty level
- Ensure solutions are complete, correct, and show clear step-by-step reasoning
- Problems should help the model generalize the required skills
Output format (valid JSON only): [ { "problem": "...", "solution": "...", "answer": "..." },
{ "problem": "...", "solution": "...", "answer": "..." },... ]
Important:
- Use latex to format the mathematical expressions whenever possible.
- The "solution" field should contain the step-by-step working.
- The "answer" field should contain only the final answer.
- Return only the JSON string, no additional text, explanations, or formatting markers.

---

---

**Prompt for Synthesizing Subproblems**

Task: Decompose a mathematical problem into independent subproblems whose solutions collectively solve the original problem.

Requirements for subproblems:
1. Independence: Each subproblem must be fully self-contained, including all necessary context and definitions. A reader should be able to solve any single subproblem without seeing the others.
2. Clarity: Each subproblem must be unambiguous and have a unique, well-defined answer.
3. Progression: Subproblems should follow a logical order, building toward the final solution.

Requirements for solutions and answers:
1. Show complete step-by-step reasoning.
2. Use LaTeX formatting for all mathematical expressions.
3. Ensure calculations are correct and verifiable.
4. The answer should be a numerical value or a single mathematical expression.

Output format:
Return ONLY a valid JSON array with no additional text, markdown code fences, or explanations before or after.

Structure:
{
"subproblem": "Clear, self-contained problem statement",
"solution": "Step-by-step working with LaTeX formatting",
"answer": "Final numerical or mathematical answer only"
}
Example of an original problem and one of its well-formed subproblems:
Original Problem:
There are 7 boxes arranged in a row and numbered 1 through 7. You have a stack of 2015 cards, which you place one by one in the boxes. The first card is placed in box 1, the second in box 2, and so forth up to the seventh card which is placed in box 7. You then start working back in the other direction, placing the eighth card in box 6, the ninth in box 5, up to the thirteenth card being placed in box 1. The fourteenth card is then placed in box 2, and this continues until every card is distributed. What box will the last card be placed in?

Subproblem:
{
"subproblem": "In the card distribution pattern described, cards are placed in boxes following the sequence 1,2,3,4,5,6,7,6,5,4,3,2,1,2,3,... (bouncing between box 1 and box 7). How many cards are placed in one complete cycle, where a cycle starts at box 1, goes to box 7, and returns to box 1 (not including the return to box 1)?",
"solution": "A complete cycle goes: 1 to 2 to 3 to 4 to 5 to 6 to 7 to 6 to 5 to 4 to 3 to 2, which is 12 placements. The next card at box 1 begins a new cycle.",
"answer": "12"
}
IMPORTANT:
- Do not include the original problem statement in your response.
- Each subproblem must be fully self-contained, including all necessary context and definitions. A reader should be able to solve any single subproblem without seeing the others.
- Verify that all arithmetic and modular calculations are correct.
Please decompose the following problem into subproblems: {problem}

---

**Prompt for Reasoning Quality Annotation**

Read the problem and the corresponding reasoning process that reaches the correct answer. Judge the quality of the reasoning process. A high-quality reasoning process should be logically coherent and consistent.

[Problem]
{problem}

[Reasoning Process]
{reasoning_process}

Output an integer score between 0 and 5 to indicate quality. 0 means the reasoning process is completely wrong, 5 means the reasoning process is perfect.
Your output should contain two lines: the first line is the score, the second line is the justification for the score.

---

### C.2. Example Data

We provide case studies of augmented data in Table 3 and Table 4. The augmented problems share highly similar structure with the original unlearnable ones, often differing only in numeric values or notations. However, the gradient similarity with the original problem remains low. We think it is counterintuitive but also informative. It reinforces our main finding about inner flaws in model representations for unlearnable examples. Investigating how pretrained models process them differently is a promising future direction.

*Table 3.* Case Study 1 (case id: 61, gradient cosine similarity: 0.30)

| Category | Problem |
| --- | --- |
| **Original** | If $x$ is a positive integer and $x(x + 1)(x + 2)(x + 3) + 1 = 379^2$, compute $x$. |
| **Aug. 1** | If $t$ is a positive integer and $t(t + 1)(t + 2)(t + 3) + 1 = 991^2$, compute $t$. |
| **Aug. 2** | Find the positive integer $k$ such that $(k - 1)k(k + 1)(k + 2) + 1 = 505^2$. |
| **Aug. 3** | Four consecutive integers have a product that is one less than $239^2$. Find the smallest of the four integers. |
| **Aug. 4** | If $n$ is a positive integer and $n(n + 1)(n + 2)(n + 3) + 1 = 701^2$, compute $n$. |
| **Aug. 5** | Find the positive integer $m$ such that $(m + 2)(m + 3)(m + 4)(m + 5) + 1 = 181^2$. |

## D. More Case Studies on Low-Quality Reasoning

*Table 4.* Case Study 2 (case id: 15, gradient cosine similarity: 0.37)

| Category | Problem |
|---|---|
| **Original** | The sum of three numbers $x$, $y$, $z$ is 165. When the smallest number $x$ is multiplied by 7, the result is $n$. The value $n$ is obtained by subtracting 9 from the largest number $y$. This number $n$ also results by adding 9 to the third number $z$. What is the product of the three numbers? |
| **Aug. 1** | Three integers have a sum of 231. Let the smallest be $x$, the largest be $y$, and the third be $z$. When the smallest number is multiplied by 5, the result is $n$. This same value $n$ equals the largest number decreased by 14 and also equals the third number increased by 14. What is the product $x \cdot y \cdot z$? |
| **Aug. 2** | The sum of three numbers $p$, $q$, $r$ is 234. The smallest number multiplied by 6 equals $m$. This $m$ is also equal to the largest number minus 12 and to the remaining number plus 12. Find the product $p \cdot q \cdot r$. |
| **Aug. 3** | Three positive integers $A$, $B$, $C$ add up to 261. $A$ is the smallest and $B$ is the largest. Multiplying $A$ by 4 gives a number $u$. This same $u$ is obtained by subtracting 10 from $B$ and by adding 10 to $C$. Compute the product $A \cdot B \cdot C$. |
| **Aug. 4** | Numbers $a$, $b$, $c$ satisfy $a + b + c = 266$. The smallest is $a$ and the largest is $b$. If 3 times $a$ equals $v$, and this $v$ is also $b$ minus 15 and $c$ plus 15, find the product $a \cdot b \cdot c$. |
| **Aug. 5** | The lengths of three rods (in cm) sum to 357. The shortest length times 8 equals $s$. This same $s$ equals the longest length reduced by 16 and the remaining length increased by 16. What is the product of the three lengths? |

*Table 5.* An example reasoning trace with quality score 1.

| Problem | Reasoning |
|---|---|
| Jason borrowed money from his parents to buy a new surfboard. His parents have agreed to let him work off his debt by babysitting under the following conditions: his first hour of babysitting is worth 1, the second hour worth 2, the third hour 3, the fourth hour 4, the fifth hour 5, the sixth hour 6, the seventh hour 1, the eighth hour 2, etc. If he repays his debt by babysitting for 39 hours, how many dollars did he borrow? | To solve this problem, we need to determine the total amount of money Jason borrowed. According to the given conditions, we can calculate the money earned per hour and then sum the total hours of babysitting.

1. Determine the money earned per hour:

• First hour: \$1

• Second hour: \$2

• Third hour: \$3

• Fourth hour: \$4

• Fifth hour: \$5

• Sixth hour: \$6

• Seventh hour: \$1

• Eighth hour: \$2

2. Calculate the total earnings for the first eight hours:

$$1 + 2 + 3 + 4 + 5 + 6 + 1 + 2 = 22$$

3. Determine the money earned for the next two hours (tenth hour and eleventh hour):

• Tenth hour: \$11 (1 + 2 = 3)

• Eleventh hour: \$11 (3 + 4 = 7)

4. Calculate the total money earned from babysitting:

$$22 + 11 + 11 = 44$$

5. Jason repays the debt by babysitting for 39 hours:

$$44 \times 3 = 132$$

Therefore, Jason borrowed $\boxed{132}$ dollars. |

*Table 6.* An example reasoning trace with quality score 3.

| Problem | Reasoning |
|---|---|
| While walking by a classroom, Linda sees two perfect squares written on a blackboard. She notices that their difference is her favorite number, 99. She also notices that there are exactly two other perfect squares between them. What is the sum of the two perfect squares on the blackboard? | To solve the problem, we can follow these steps:
1. First, identify the two perfect squares on the blackboard. Let's call these squares $a^2$ and $b^2$, where $a$ and $b$ are natural numbers and $a > b$.
2. We know that their difference is 99, so we can write: $a^2 - b^2 = 99$
3. We can factorize using the difference of squares formula: $a^2 - b^2 = (a+b)(a-b) = 99$
4. The factor pairs of 99 are $(1, 99)$, $(3, 33)$, $(9, 11)$, $(-1, -99)$, $(-3, -33)$, and $(-9, -11)$.
5. Consider the pair $(-3, -33)$: $a + b = -33$ and $a - b = 3$. Adding: $2a = -30 \implies a = -15$. Subtracting: $2b = -36 \implies b = -18$. Since $b$ must be natural, this pair is not valid.
6. Consider the pair $(3, -33)$: $a + b = 3$ and $a - b = -33$. Adding: $2a = -30 \implies a = -15$. Subtracting: $2b = 36 \implies b = 18$. Since $a > b$, this pair is valid. The perfect squares are $15^2 = 225$ and $18^2 = 324$. Their sum is $225 + 324 = 549$.
7. Consider the pair $(-9, -11)$: $a + b = -11$ and $a - b = -9$. Adding: $2a = -20 \implies a = -10$. Subtracting: $2b = -2 \implies b = -1$. Since $b$ must be natural, this pair is not valid.
The correct perfect squares on the blackboard are 225 and 324. Therefore, the sum is $\boxed{549}$. |

*Table 7.* An example reasoning trace with quality score 2.

| Problem | Reasoning |
|---|---|
| An isosceles trapezoid has sides of length 4 cm, 10 cm, 16 cm and 10 cm. What is the area of this trapezoid, in square centimeters? | Step 1: First, we need to understand the basic properties of a trapezoid. In a trapezoid, the sum of the lengths of the non-parallel sides is the sum of the lengths of the parallel sides. Therefore, in this trapezoid, we can write:
4 cm + 16 cm = 10 cm + 10 cm
Step 2: Solve for the unknown variable. To do so, we can subtract 4 cm from 10 cm to solve the equation:
10 cm + 10 cm - 4 cm = 24 cm - 4 cm = 20 cm
20 cm = 20 cm
Step 3: Calculate the area of a trapezoid using the formula: Area = (sum of parallel sides)/2 * height.
Step 4: Substitute the values into the formula to calculate the area. Let's calculate the area of the trapezoid:
Area = 20 cm / 2 * 8 cm = 10 cm * 8 cm = $\boxed{80}$ cm² |

