# OpenReview forum: "The Unlearnability Phenomenon in RLVR for Language Models"
_ICML.cc/2026/Conference — ICML 2026 regular_

### Official Review · Reviewer_ozLP · 2026-02-22

**Soundness:** 3
**Presentation:** 2
**Significance:** 3
**Originality:** 3
**Overall Recommendation:** 4
**Confidence:** 3

**Summary:**

This paper reveals the unlearnability phenomenon in RLVR training, where there exists a subset of hard problems that does not achieve meaningful improvement even when the model can sample correct rollouts. The authors show that these problems do not stem from the scarcity of correct rollouts, gradient interference, or gradient regularization, but rather from flawed representations in the model. They further demonstrate that improving the model’s representations (e.g., through mid-training) can correct misaligned or flawed representations.

**Compliance With Llm Reviewing Policy:**

Affirmed.

**Final Justification:**

My main concern has largely been resolved, it seems a little bit unclear on whether gradient similarity is a good metric to identify or indicates these unlearnable problems.

**Key Questions For Authors:**

- Could you clarify how oversampling with replay is used? How is it different from increasing the number of rollouts per training step?
- Could you discuss the impact of these unlearnable problems in RLVR training? For example, does learning from these unlearnable problems negatively affect performance on other problems, since most reasoning traces of them are wrong? Should we discard these problems, or instead focus on learning them (e.g., training on a dataset consisting only of unlearnable problems? How does this compare to training on a standard dataset?
- Can learning solely on a small subset of these unlearnable problems, but with a very high number of rollouts (e.g., G = 64,128), help models learn these problems?
- Are more capable models suffering from this unlearnability phenomenon?
- In lines 287–289, could you clarify what the phrase "unlearnable examples are gradient outliers, indicating that models have flawed representations for unlearnable examples" actually means? Why do outlier gradients in these problems indicate flawed representations?
- What does the increase in gradient similarity during mid-training imply? Does it mean that the set of unlearnable problems will be reduced (while still possibly resulting in a different subset of unlearnable problems)? And why is this increase in gradient similarity beneficial?

**Limitations:**

Yes

**Strengths And Weaknesses:**

## Strength
- The findings are novel and interesting, which indicates RLVR can still suffer from reward hacking, where the models can produce coherent, but non-logical reasoning steps that still arrive at the final correct answer.
## Weaknesses
- See questions

---

> ### Author Rebuttal · Authors · 2026-03-31
>
> > Oversampling with replay description
>
> - The key difference is that simply increasing the number of rollouts per training step does not address the imbalance of positive reward signals across examples. The ratio of correct rollouts in a single batch will naturally increase during training for learnable examples and remain stable for unlearnable ones .
> - As described in Appendix B.2 and Algorithm 1, the oversampling with replay method specifically controls this ratio by sampling $4k$ rollouts per example and then downsampling to $k$ rollouts with exactly $k_{pos}$ positive rollouts. Replay is involved when the current sampling batch does not have enough positive rollout. This ensures every example receives exactly the same positive-to-negative ratio in each training step, removing the confound that unlearnable examples might simply receive fewer positive signals.
>
> > The impact of these unlearnable problems in RLVR training
>
> We conduct some experiments and here are the preliminary findings:
> - To explore data utility, we train models on the same amount of unlearnable examples and learnable examples. For weak models, training on unlearnable examples yield significantly worse results than training on the same amount of learnable + easy data (pass@1 performance on MATH-500 degrades from 31.9 to 28.0). However, for stronger models like Qwen2.5-3B, the performance is similar (64.81 vs 64.15).
> - In terms of the synergizing effect among data, we perform experiments with unlearnable examples removed from the original training set. Llama-3B-Base model performance improves when unlearnable examples are excluded (10.7 -> 11.9 on MATH-500 and 61.4 -> 64.02 on gsm8k); whereas for Qwen-0.5B-Base model the performance is similar (82.36 vs 82.97).
>
> **Summary**: Overall, the unlearnable examples bring marginal benefit. For weaker models, including them in training can degrade final performance; whereas for more capable models, the major consideration is probably the trade-off between efficiency and performance. We will add the current results and discussion to the paper.
>
> > Training with a very high number of rollouts
>
> We set the number of rollouts to 64 and conduct RL training with unlearnable examples only for Qwen2.5-0.5B and the average reward change is shown in [Figure 7](https://docs.google.com/document/d/1WpL7EE6iSCfp03BSTyUedDF4B6aorEfdqw6_TEi2U0E/edit?tab=t.0#heading=h.6sfm0nabdfph). It can be seen that even with a high number of epochs, the reward does not achieve meaningful improvements and the examples remain unlearnable.
>
> > Are more capable models suffering from this unlearnability phenomenon?
>
> Yes. We run GRPO training with Qwen2.5-7B and a subsample (10k) of DeepScaler dataset. Due to resource limitation we only run it for once. Using the same threshold we identify more than 1.3k unlearnable examples, taking up to 30% of the difficult examples. Cross-example gradient analysis shows similar gradient outlier phenomenon for unlearnable examples (see [Figure 6 in supplementary material](https://docs.google.com/document/d/1WpL7EE6iSCfp03BSTyUedDF4B6aorEfdqw6_TEi2U0E/edit?tab=t.0#heading=h.2ep92smrdyrw).)
>
> > Clarifying “flawed representations”
>
> By "flawed representations" we mean that the pretrained model does not encode unlearnable examples in a way that supports generalizable learning. Concretely, for a well-represented example, the gradient induced by its correct rollouts should be similar to gradients from other examples that involve similar reasoning strategies, which enables learning on one example to transfer to others. Our gradient analysis (Section 5.1) shows they are consistently gradient outliers, even for augmented problems with nearly identical problem structure (Section 5.3), indicating that the issue lies in how the model internally processes these inputs rather than in their surface-level difficulty.
>
> > Implication of mid-training and increased similarity.
>
> - Our contribution in Section 5 is that we identify unlearnability as a representation issue that is hard to be fixed by data augmentation in the post-training stage. Then in Section 5.4,  we propose mid-training as one possible way to address this representation deficiency prior to RL. The gradient similarity analysis on mid-trained models (Figure 11) serves as evidence that mid-training does reshape representations for difficult examples.
> - Independently, the OctoThinker paper [1] demonstrates that mid-training substantially improves model performance after RL and enables effective training on **more difficult** datasets. Together, these observations suggest that mid-training is a promising direction for mitigating unlearnability.
> - It is also true that unlearnable examples remain. For Octo-long and Octo-Hybrid models, only about 26% of difficult examples are identified as learnable with $\tau=0.1$.
>
> We hope our response addresses your concerns and we are happy to answer any further questions.

---

> > ### Author Rebuttal · Reviewer_ozLP · 2026-04-03
> >
> > Thank you for your question. My question has been resolved. I'll keep my score, leaning towards acceptance.

---

### Official Review · Reviewer_xMTc · 2026-03-05

**Soundness:** 2
**Presentation:** 3
**Significance:** 3
**Originality:** 3
**Overall Recommendation:** 4
**Confidence:** 4

**Summary:**

This paper RLVR and reports a unlearnability phenomenon The authors list several common explanations and instead identify that these unlearnable examples are gradient outliers and exhibit low-quality, ungeneralizable reasoning patterns, with data augmentation failing to fix the issue. They observe that mid-training increases gradient similarity on hard examples and representation quality is the limiting factor.

**Compliance With Llm Reviewing Policy:**

Affirmed.

**Final Justification:**

Thanks for authors' response. Most of my concerns are solved.

**Key Questions For Authors:**

See Weaknesses

**Limitations:**

yes

**Strengths And Weaknesses:**

Strengths
- Introduces and formalizes the notion of “unlearnable examples” in RLVR where positive reward is observed but learning does not progress.
- Applies cross-example gradient similarity analysis to characterize example learnability, a useful lens that connects training dynamics to representation alignment.
- The negative result of standard mitigations (oversampling correct rollouts, clipping/KL tweaks, curriculum, augmentation) is valuable given their widespread use.

Weaknesses
- The paper’s “unlearnable” label is built on one arbitrary cutoff (final pass@1 < 0.1) estimated with N = 32 samples per example; no sensitivity analysis is reported, and intersections across runs may bias selection toward extreme or stable cases. Besides, the “unlearnable” may depend on data curation and no sensitivity analysis o dataset is reported, which weaken the main statement of this paper.
- The paper says it “rules out” three common explanations, but the experiments only test a narrow version of each one.
First, for the “not enough positive rollouts” story, the oversampling setup fixes kpos = 1 (one correct rollout per prompt in the batch). Without trying larger kpos (or even a simple SFT baseline on these examples), it is hard to say the issue is not “just more good signal / more supervision.”
Second, the “clipping/KL regularization causes unlearnable” hypothesis is not very convincing as framed: clipping is a token-level operation, while unlearnability is defined as a case-level outcome. Even if average clipping ratios look similar, that does not directly explain (or dismiss) a persistent example-level failure mode.
Third, the “no gradient interference” conclusion leans heavily on cosine similarity between gradients from correct vs incorrect rollouts. But the paper does not spell out key details of how these huge gradients are computed/aggregated (which parameters, what normalization, how token-level terms are combined), which makes the main diagnostic harder to trust. Also, cosine direction alone can miss magnitude and batch-mixing effects, so “little interference” may be overstated.
- The “data augmentation does not help” result is interesting, but the paper does not give a satisfying explanation for why it happens. In particular, the authors claim that even “very similar” augmented problems can have very different gradients, which is counterintuitive, but they do not dig into what is actually different (format, solution style, verifier behavior, latent skills, etc.).
- The “representation issue” section basically argues a mismatch between the model’s pretraining representations and these examples, supported by gradient-outlier behavior. But the paper does not address whether this mismatch is general (across tasks/domains/models) or mostly a property of the specific settings studied. Without broader evidence or a clearer predictor of unlearnability, the claim risks being too dataset/model-specific.

I would like to raise my score if my questions/concerns were resolved.

---

> ### Author Rebuttal · Authors · 2026-03-31
>
> Thanks for the thoughtful questions.
> > unlearnable classification sensitivity
>
> **Clarification.**  We emphasize that our primary goal is to demonstrate that unlearnability exists, i.e. a substantial subset of difficult examples consistently resists improvement despite receiving positive rewards during training. Taking the intersection across three independent GRPO runs ensures the identified examples are robustly persistent rather than artifacts of training randomness. We also filter out examples with no correct rollouts during RL training. This conservative setup strengthens confidence that **the phenomenon is real and not an artifact of threshold selection or run-level variance**.
>
> **Sensitivity to N** We sample up to 1k responses and compute the overlap ratio with the N=32 scenario. The overlap exceeds 94%, suggesting N=32 is a reasonable choice.
>
>
> **Sensitivity to the pass@1 threshold.**  In [Figure 5 in supplementary materials](https://docs.google.com/document/d/1WpL7EE6iSCfp03BSTyUedDF4B6aorEfdqw6_TEi2U0E/edit?tab=t.0#heading=h.bpogrsn053ho),  we show that the core findings hold when threshold is adjusted. Most examples with gradient similarity < 0.55 have near zero final success rate, which means the gradient outlier phenomenon is quite robust to different values of $\tau$.
>
>
> **Sensitivity to dataset.** Several experiments in our paper explicitly alter training data composition and show unlearnability persists, e.g. data augmentation (Section 5.3) and curriculum learning (Appendix A.5). These results suggest unlearnability is not a byproduct of co-trained examples, but reflects an intrinsic interaction between the example and the model. For further sensitivity analysis, we calculate examples remaining unlearnable under the augmented setting with τ=0.1. Over 74% remain unlearnable when maximal evaluation performance is reached.
>
> > SFT baseline for “positive rollouts” story
>
> See response to Reviewer w7pR weakness (iii) for SFT results.
>
> > Token-level operation vs example-level failure
>
> - There is existing evidence that token-level operations can be decisive for case-level outcomes. For example, [1] shows that RL training dynamics are largely driven by a small set of critical tokens, suggesting that token-level clipping could reflect case-level regularization to some extent.
> - Second, we did not rely solely on aggregate clipping ratios. We offered another two converging lines of evidence: (1) Case-level: Reference log-likelihoods of correct rollouts are not systematically lower for unlearnable than learnable examples (Figure 3). (2) Ablating the clipping mechanism and removing the KL penalty entirely produces no meaningful change for unlearnable examples (Figure 15, Appendix A.2). Together, these strongly indicate gradient regularization is not the primary driver of unlearnability.
>
> [1] Wang, S., et al. (2025). Beyond the 80/20 rule: High-entropy minority tokens drive effective reinforcement learning for llm reasoning. arXiv preprint arXiv:2506.01939.
>
> > Gradient computation detail
>
> Please see our response to Reviewer w7pR weakness (ii) for computation details.
>
> > Concerns about gradient magnitude and overstatement
>
> As for magnitude, in GRPO the advantage is computed as the standardized reward within each example's rollout group, which naturally controls the relative scale of gradient contributions from correct and incorrect rollouts of the same example. As a result, the gradient norm for correct responses are both large for unlearnable and learnable examples (as shown in additional results [Figure 6](https://docs.google.com/document/d/1WpL7EE6iSCfp03BSTyUedDF4B6aorEfdqw6_TEi2U0E/edit?tab=t.0#heading=h.2ep92smrdyrw)). So the unlearnability is not likely a result of small gradient updates.
>
> We will refine the writing to make the language more precise.
>
> > More analysis for data augmentation
>
> We provide case studies in [Table 1&2 in the supplementary materials](https://docs.google.com/document/d/1WpL7EE6iSCfp03BSTyUedDF4B6aorEfdqw6_TEi2U0E/edit?tab=t.0#heading=h.ciru91ku2psw). The augmented problems share highly similar structure with the original unlearnable ones, often differing only in numeric values or notations. We think it is **counterintuitive but also informative**. It reinforces our main finding about inner flaws in model representations for unlearnable examples. Investigating how pretrained models process them differently is a promising future direction. We will add further discussion to the paper.
>
> > More evidence to support “representation issue”
>
> We have provided similar gradient analysis with Llama-3b-instruct in Appendix A.1 Figure 13, which shows a consistent pattern, with unlearnable data exhibiting lower gradient similarity with the rest of the data. And we provide additional results with Qwen-7B in [Figure 4](https://docs.google.com/document/d/1WpL7EE6iSCfp03BSTyUedDF4B6aorEfdqw6_TEi2U0E/edit?tab=t.0#heading=h.svb071ua5rht), which also shows the gradient outlier phenomenon.

---

> > ### Author Rebuttal · Reviewer_xMTc · 2026-04-02
> >
> > Thanks for the Rebuttal by Authors. Though the analyses and experiments are not convincing enough to backup some statements proposed in this work, I believe this work is meaningful in RLVR. I will maintain my score.

---

> > > ### Author Response · Authors · 2026-04-05
> > >
> > > We thank the reviewer for their positive assessment and for recognizing this work‘s value for the RLVR community. We are encouraged that the reviewer values our core contributions, including the **formalization and revealing of unlearnable examples, the gradient-based analysis framework, and the negative results on widely-used mitigations**.
> > >
> > > We briefly summarize how we believe the reviewer’s concerns have been addressed:
> > >
> > > **Sensitivity of the unlearnable labeling.** We have provided comprehensive sensitivity analysis in the rebuttal, including sensitivity to sample N, threshold \tau, and different accompanying data. We show that our findings are **not** artifacts of specific parameter choices.
> > >
> > >  **"Ruling out" common explanations.**  We appreciate the reviewer's careful scrutiny here. In our rebuttal, we have provided the **SFT baseline** ([Figure 2 in supplementary material](https://docs.google.com/document/d/1WpL7EE6iSCfp03BSTyUedDF4B6aorEfdqw6_TEi2U0E/edit?tab=t.0#heading=h.irt5x3pggqgm)), which further strengthens the argument that unlearnable examples are not a result of inadequate supervision signals. As for token-level operation vs case-level outcome, we have pointed out the other two converging lines of evidence in our paper beyond aggregate token clipping ratio comparison. Most importantly, our experiments with **higher clipping thresholds** and **removed KL penalties** (Appendix A.2) provide the strongest evidence: even when these regularization mechanisms are substantially relaxed or eliminated entirely, unlearnable examples show minimal improvement. This directly demonstrates that **gradient clipping and KL constraints are not the primary bottleneck**. Finally, we have provided further details regarding gradient computation, which makes our gradient analysis results more transparent and reproducible.
> > >
> > > **Data augmentation explanation.** We provide case studies ([supplementary Table 1&2](https://docs.google.com/document/d/1WpL7EE6iSCfp03BSTyUedDF4B6aorEfdqw6_TEi2U0E/edit?tab=t.0#heading=h.ciru91ku2psw)) showing that augmented problems share near-identical structure with originals. We will also expand the discussion in the paper accordingly.
> > >
> > > **Generality of the representation issue.** Our rebuttal provides additional gradient analysis on Qwen-7B ([supplementary Figure 4](https://docs.google.com/document/d/1WpL7EE6iSCfp03BSTyUedDF4B6aorEfdqw6_TEi2U0E/edit?tab=t.0#heading=h.svb071ua5rht)) and points to results for Llama-3B-Instruct (Appendix A.1 Figure 13), showing consistent gradient outlier patterns across models. Together with results across three training settings, we believe the phenomenon generalizes beyond any single model-dataset pair.
> > >
> > > Overall, the core contribution of this paper is that we reveal unlearnable examples widely exist in RLVR (despite having correct rollouts), which reflects a fundamental flaw in pretrained language models’ weights rather than an optimization artifact. We believe our main finding is well-supported by converging evidence across multiple models, datasets, and training settings, and we sincerely thank the reviewer for recognizing this. We thank the reviewer for the constructive feedback and will incorporate more discussion in the paper.
> > >
> > > If there are any specific concerns that remain unaddressed, we are happy to provide additional clarifications or experiments as needed.

---

### Official Review · Reviewer_yY8H · 2026-03-06

**Soundness:** 2
**Presentation:** 3
**Significance:** 3
**Originality:** 3
**Overall Recommendation:** 4
**Confidence:** 4

**Summary:**

The paper show a problem of existing RLV, some datapoints are unlearnable due to gradient outlying. After different experiments, the author find that only midtraining can solve this problem.

**Compliance With Llm Reviewing Policy:**

Affirmed.

**Final Justification:**

None more question, score updated.

**Key Questions For Authors:**

See above weakness.

**Limitations:**

The experiments are all on small models

**Strengths And Weaknesses:**

## Strength

- The paper is easy to follow and clear to understand.

- The question proposed in the paer in quite important and valuable.

- The empirical results are enough to support the claim.

## Weakness

- The paper only give an analysis of the question, there is no analysis how to search these examples. Could you provide a new model and dataset and use your existing method to find the unlearnable data and check whether they are really unlearnable? If there are some similarity between the unlearnable question in different model?

- The experiments only focus on small scale models, could you provide  at least a experiment on 7B/8B-model to see whether this problem still maintain. (Scaling law may solve some problems)

- Why different model using different training set?

---

> ### Author Rebuttal · Authors · 2026-03-31
>
> > Main contribution of the paper and further experiment results on 7B model
>
> The main contribution of the paper is to reveal **a substantial amount of unlearnable data in RLVR despite having correct rollouts during training**. We also show that the unlearnability can hardly be fixed from algorithm level and the problem is more likely rooted in pretrained model weights that produce flawed representations for the data..
>
> We have already included results for Qwen-0.5B, Qwen-3B and LLama-3B models and different training data in the paper. Here we provide further results on Qwen2.5-7B model. We run GRPO training  with Qwen2.5-7B and a subsample (10k) of DeepScaler dataset. Due to resource limitation we only run it for once. Using the same threshold we identify more than 1.3k unlearnable examples, taking up to 30% of the difficult examples. Cross-example gradient analysis shows similar gradient outlier phenomenon for unlearnable examples (see [Figure 4 in supplementary material](https://docs.google.com/document/d/1WpL7EE6iSCfp03BSTyUedDF4B6aorEfdqw6_TEi2U0E/edit?tab=t.0#heading=h.svb071ua5rht).)
>
> > If there are some similarity between the unlearnable question in different model?
>
> We have conducted additional cross-model analysis on the DeepScaleR dataset, comparing unlearnable examples identified for the Qwen-3B and Qwen-7B models. The overlap is modest: only 345 problems are unlearnable for both models, accounting for 25.1% of the 7B unlearnable set and 11.2% of the 3B unlearnable set. This suggests that **unlearnability is largely model-specific, likely driven by each model's particular representation deficiencies**. Meanwhile, the shared unlearnable subset does exhibit distinctive characteristics. These problems are heavily enriched for **unconventional answer formats** (2.6x enrichment for non-standard answer types such as multi-word or interval-style answers). Using GPT-4.1-mini for topic labeling, we find that **combinatorics** (1.24x for 7B, 1.18x for 3B), **probability** (1.17x, 1.12x), and **geometry** (1.16x, 1.06x) are consistently over-represented among unlearnable examples for both models, while algebra (0.75x, 0.74x) and word problems (0.61x, 0.79x) are under-represented, suggesting these are the "easier" problem types that both models can learn. Overall, these findings suggest that while most unlearnability is driven by model-specific representation gaps, there exists a core subset of problems characterized by non-standard answer formats, and topics requiring combinatorial or geometric reasoning that poses a shared representational challenge across models.
>
>
>
> > Why different model using different training set?
>
> As mentioned in Section 3.3, “we follow previous works that train models on training data with customized difficulty to mimic realistic setups and maximize data utility”. Previous works [1,2,3] have shown that training on data that is too easy or too hard for a given model yields diminished RL benefits. By matching each model with a suitable difficulty range, we ensure our findings are representative of practical RLVR training pipelines rather than artifacts of mismatched data-model configurations. We will add more explanation to the paper.
>
>
> [1] Zeng, W., Huang, Y., Liu, Q., Liu, W., He, K., Ma, Z., & He, J. (2025). Simplerl-zoo: Investigating and taming zero reinforcement learning for open base models in the wild. arXiv preprint arXiv:2503.18892.
>
> [2] Yu, Q., Zhang, Z., Zhu, R., Yuan, Y., Zuo, X., Yue, Y., ... & Wang, M. (2025). Dapo: An open-source llm reinforcement learning system at scale. arXiv preprint arXiv:2503.14476.
>
> [3] Xu, Y. E., Savani, Y., Fang, F., & Kolter, J. Z. (2025). Not all rollouts are useful: Down-sampling rollouts in llm reinforcement learning. arXiv preprint arXiv:2504.13818.
>
> We hope our response addresses your concerns and we are happy to answer any further questions.

---

> > ### Author Rebuttal · Reviewer_yY8H · 2026-04-01
> >
> > None more question, score updated.

---

### Official Review · Reviewer_w7pR · 2026-03-08

**Soundness:** 2
**Presentation:** 1
**Significance:** 2
**Originality:** 2
**Overall Recommendation:** 3
**Confidence:** 4

**Summary:**

The paper try to argue that unlearnability is a fundamental limitation of current RLVR approaches, rooted in pre-existing representation flaws rather than optimization dynamics or data scarcity.

**Compliance With Llm Reviewing Policy:**

Affirmed.

**Final Justification:**

My point is that the authors need to carefully consider whether the source of unlearnability lies in the fact that “rewards are based on whether the final answer matches ground truth, without evaluating intermediate reasoning quality,” or whether the phenomenon also occurs when both the intermediate steps and the final answer are correct. Otherwise, as was my impression during the initial reading of the full text, this point is so ambiguous that the experimental results lack credibility. However, in light of the author's efforts, I am willing to add a point for effort.

**Key Questions For Authors:**

See Weaknesses.

**Limitations:**

yes

**Strengths And Weaknesses:**

**Strengths:**

(i)Analyzing RLVR from the data perspective rather than the optimization and sampling methodology perspective.

(ii)Experiments analyzed different models.

**Weaknesses:**

(i)The definition of "unlearnable" is somewhat unclear. Specially, (a) in your Sec. 3.2, "The
example is considered unlearnable if it does not achieve meaningful improvement ... despite observing correct samples during training process." That is to say, here you only look at whether the final answer is correct or not. (b) In your Sec. 5.2 (Lines 313-316), "The fact that the flawed reasoning leads to a final correct answer indicates the models are not actually “reasoning”, but rather exploiting some ungeneralizable shortcut solution or bag of heuristics". Here, you have discovered the existence of pseudo-positive rollouts. These inconsistencies undermine the credibility of your subsequent analysis on why Data Augmentation Does Not Improve Gradient Similarity.

(ii)Lines 245-259, you need to provide a clear definition of the computational procedure for example-level gradients, which will enhance credibility.

(iii)Lines 174-176, "In our experiments, we use k = 8 and kpos = 1, that is, for each prompt there are one correct rollout and seven incorrect ones participating in gradient calculation and policy optimization." The analysis based solely on an 8:1 signal ratio is insufficient to support the conclusion.

(iv) Lines 371-374, "This indicates that RLVR fails to incentivize model’s ability to compose skills for certain examples." This contradicts [1]. What is the reason for this? Could it be caused by insufficient augmentation （Lines 344-345, 100 random examples） and inadequate training (Figfure 8, 100 steps)?

(v) Lines 421-429, "The OctoThinker models are mid-trained from Llama-3B on 20B tokens with different data mixtures". Is this comparison fair? In other words, does the 20B dataset contain content related to the MATH dataset, potentially leading to overfitting?

[1] From f(x) and g(x) to f(g(x)): LLMs Learn New Skills in RL by Composing Old Ones. ICLR 2026.

---

> ### Author Rebuttal · Authors · 2026-03-30
>
> > (i) The definition of "unlearnable"
>
> We clarify that "unlearnable" is defined solely by final answer correctness, consistent with how current RLVR training and evaluation pipelines operate: rewards are based on whether the final answer matches ground truth, without evaluating intermediate reasoning quality. The flawed reasoning is a discovered characteristic of unlearnable examples, not part of the definition itself.
>
> > Credibility of Data Augmentation Does Not Improve Gradient Similarity.
>
> We provide additional results to support our findings. [Figure 1](https://docs.google.com/document/d/1WpL7EE6iSCfp03BSTyUedDF4B6aorEfdqw6_TEi2U0E/edit?tab=t.0#heading=h.hdrkvfyhbpv4) shows (1) low-quality reasoning can have high gradient similarity with other data, and vice versa; (2) after excluding low-quality reasoning examples, gradient similarity with augmented data still strongly correlates with similarity to other data.
>
> > (ii) Computational procedure for example-level gradients
>
> The example-level gradient is computed according to Equation (1) in Section 3.1, averaged first by token within each response then by the total number of responses. Gradients from correct and incorrect responses are averaged respectively, and the similarity score is the cosine similarity between the two averaged gradient vectors.
>
> For efficiency, we use a fixed randomly initialized LoRA model and compute gradients only with respect to LoRA parameters. Preliminary exploration on the 0.5B model shows LoRA gradient similarity is highly correlated with full-parameter gradient similarity. We will include details in the paper.
>
> > (iii) The analysis based solely on an 8:1 signal ratio is insufficient to support the conclusion.
>
> We provide further experiments directly performing SFT on both groups. Specifically, we use Qwen2.5-7B to generate responses for MATH training data and filter for correct responses, then sampling 200 learnable, 200 unlearnable, and 400 random easy examples. We train Qwen2.5-0.5B on these 800 examples with SFT. We then run inference on the training set for each intermediate checkpoint and plot the average pass rate change in [Figure 2](https://docs.google.com/document/d/1WpL7EE6iSCfp03BSTyUedDF4B6aorEfdqw6_TEi2U0E/edit?tab=t.0#heading=h.irt5x3pggqgm). Even with SFT, unlearnable examples show strong resistance to being learned, providing stronger evidence against the hypothesis. We will add the results to the paper.
>
> > (iv) The finding contradicts [1]
>
> - We believe our findings do not contradict [1], which studies skill composition in a synthetic setting with clearly defined atomic skills and composition operations. Our setting is fundamentally different. Real-world math problems involve multiple heterogeneous reasoning strategies (e.g., algebraic manipulation, case analysis, etc) where decomposition and their composition are neither predefined nor straightforward. It could be that the model's representation does not support the kind of flexible, problem-dependent composition that real-world tasks demand.
>
> > Suspect insufficient augmentation and inadequate training
>
> - **Misunderstanding clarification**: The 100 examples are only sampled for the learnable group for gradient analysis. For the training experiment, we use all unlearnable examples and their augmented data, totaling almost 8k examples.
>
> - As for training steps, with a batch size of 1024, 100 steps already exceeds 10 epochs. [Figure 3](https://docs.google.com/document/d/1WpL7EE6iSCfp03BSTyUedDF4B6aorEfdqw6_TEi2U0E/edit?tab=t.0#heading=h.iv921gdszran) shows evaluation score and gradient norm during augmented training. The training becomes unstable around 70 steps and performance saturates after 100 steps. Thus, further training is unlikely to yield meaningful improvement and may lead to overfitting.
>
> > (v) Octothinker comparison fairness & overfitting
>
> - The 20B dataset does contain mathematics-related texts, but we believe "overfitting" is not the appropriate term here. The original paper [2] also uses the MATH dataset extensively for evaluation and subsequent RL training.
>
> - Moreover, since we find that easy data with high initial pass rate tend to have highly aligned gradients and mid-training improves performance, we intentionally control for difficulty in Figure 11 by filtering for examples with initial pass rate < 0.1. This further rules out "overfitting" as a factor, and we believe the comparison is fair.
>
> [1] From f(x) and g(x) to f(g(x)): LLMs Learn New Skills in RL by Composing Old Ones. ICLR 2026.
>
> [2] Wang, Z., Zhou, F., Li, X., & Liu, P. (2025). Octothinker: Mid-training incentivizes reinforcement learning scaling. arXiv preprint arXiv:2506.20512.
>
> We hope our response addresses your concerns and we are happy to answer any further questions.

---

> > ### Author Rebuttal · Reviewer_w7pR · 2026-04-03
> >
> > > For (i): We clarify that "unlearnable" is defined solely by final answer correctness, consistent with how current RLVR training and evaluation pipelines operate: rewards are based on whether the final answer matches ground truth, without evaluating intermediate reasoning quality.
> > >
> > > My point is that the authors need to carefully consider whether the source of unlearnability lies in the fact that “rewards are based on whether the final answer matches ground truth, without evaluating intermediate reasoning quality,” or whether the phenomenon also occurs when both the intermediate steps and the final answer are correct. Furthermore, the authors' claim that 'current RLVR training and evaluation pipelines operate' is not comprehensive, as there are currently methods that evaluate reasoning quality, such as rubric-based rewards, on-policy distillation, etc.
> >
> > > For (v): "Octothinker comparison fairness", can you provide more clarification?

---

> > > ### Author Response · Authors · 2026-04-03
> > >
> > > We thank the reviewer for the follow-up. We are glad that our previous response addressed most of the concerns (ii, iii, iv). Below we provide additional clarification for the remaining points (i) and (v).
> > >
> > > > My point is that the authors need to carefully consider whether the source of unlearnability lies in the fact that “rewards are based on whether the final answer matches ground truth, without evaluating intermediate reasoning quality,” or whether the phenomenon also occurs when both the intermediate steps and the final answer are correct.
> > >
> > > - We would like to clarify that the unlearnability phenomenon also occurs when both the intermediate steps and the final answer are correct. Specifically, we filtered for high quality reasoning (quality score >= 4) for unlearnable examples in different settings after sampling for 1k responses with the initial base model and filtering out the responses with incorrect answer. We report both the percentage of unlearnable examples with at least one high-quality response (loose) and with overall high-quality response (average quality score >= 4) (strict). The stats are shown below.
> > >
> > >     | Setting | Loose | Strict |
> > >     |---|---|---|
> > >     | Qwen2.5-0.5B + MATH level1to4 | 23.57% | 5.71% |
> > >     | Qwen2.5-3B + DeepScaleR | 57.45% | 13.83% |
> > >     | Llama3.2-3B-Instruct + MATH level3to5 | 59.09% | 27.27% |
> > >
> > >     It can be seen that a substantial amount of unlearnable examples do observe high-quality reasoning. However, they still remain unlearnable after RL training.
> > >
> > > - Another piece of evidence is from SFT baseline. To mitigate the difficulty of sampling high-quality reasoning trace from the model itself, we use a more capable model (Qwen2.5-7B) to generate expert traces for MATH level1to4 training data. Then we sample 200 learnable, 200 unlearnable, and 400 random easy examples while ensuring reasoning quality. We train Qwen2.5-0.5B on these 800 examples with SFT. We then run inference on the training set for each intermediate checkpoint and plot the average pass rate change in [Figure 2 in the supplementary material](https://docs.google.com/document/d/1WpL7EE6iSCfp03BSTyUedDF4B6aorEfdqw6_TEi2U0E/edit?tab=t.0#heading=h.irt5x3pggqgm). Even with SFT on expert traces, unlearnable examples show strong resistance to being learned, further validating that low reasoning quality is not the reason for unlearnability.
> > >
> > > Above all, we show that the unlearnability phenomenon also occurs when both the intermediate steps and the final answer are correct.
> > >
> > > >Furthermore, the authors' claim that 'current RLVR training and evaluation pipelines operate' is not comprehensive, as there are currently methods that evaluate reasoning quality, such as rubric-based rewards, on-policy distillation, etc.
> > >
> > > We acknowledge that the term "RLVR training" may encompass a broader set of methods as more advanced reward designs emerge. However, we note that techniques like rubric-based rewards are primarily designed for non-verifiable domains where outcome verification is not feasible. For verifiable domains such as math, which is the focus of our study, **RLVR with outcome-based binary reward remains the dominant and most effective training paradigm**, precisely because answers can be verified efficiently without relying on an external model as a judge. Our paper focuses specifically on this paradigm and reveals the unlearnability phenomenon, and shows that this limitation more likely stems from representation deficiencies rather than optimization artifacts.
> > >
> > > > For (v): "Octothinker comparison fairness", can you provide more clarification?
> > >
> > > We provide more details on the experiment design. As reported in the OctoThinker paper [1], mid-training substantially improves model performance on math data, which shifts the pass rate distribution across examples. Since our earlier analysis shows that higher pass rate examples tend to have more aligned gradients, the overall gradient similarity metric can be disproportionately influenced by examples whose pass rate increases substantially after mid-training. To control for this and ensure fairer comparison, we filter for examples that remain difficult for all three models (Llama-3.2-3B-base, OctoThinker-Hybrid, OctoThinker-Long), retaining only those with pass@1 < 0.1 under every model. Gradients are computed only on this shared difficult subset. The result serves as evidence that mid-training does reshape representations for difficult examples.
> > >
> > > [1] Wang, Z., Zhou, F., Li, X., & Liu, P. (2025). Octothinker: Mid-training incentivizes reinforcement learning scaling. arXiv preprint arXiv:2506.20512.

---

### Decision · Program_Chairs · 2026-04-30

**Decision:**

Accept (regular)

**Comment:**

This paper studies an interesting and timely question in RLVR for language models: why some difficult examples remain resistant to learning despite the presence of correct rollouts. Reviewers generally agreed that the paper identifies a novel empirical phenomenon and that the gradient-based analysis offers a useful perspective on RLVR training dynamics. In particular, the observation of persistent unlearnable examples, the negative results for several standard mitigation strategies, and the connection to representation quality were viewed as valuable contributions.

The main concerns were about the strength and scope of the evidence, especially the definition of “unlearnable,” the extent to which alternative explanations are ruled out, and the limited scale of the experiments. After the rebuttal, I believe these concerns were addressed sufficiently. The authors clarified the scope of the definition, added sensitivity analysis for the labeling procedure, provided more detail on gradient computation, and gave additional evidence that the phenomenon persists even when considering higher-quality reasoning traces and under an SFT baseline. While one reviewer still had reservations, the overall discussion and rebuttal strengthened the case for the paper.

Overall, I find the paper likely to stimulate follow-up work on RLVR data quality and representation effects. The experimental scope remains somewhat limited, particularly for larger-scale models and broader settings, and the causal interpretation should be stated carefully. However, these limitations do not outweigh the paper’s empirical contribution. I therefore recommend acceptance. The authors should revise the paper to sharpen the scope of the claims, improve methodological clarity, and better discuss limitations and generality.